# On the Notion of Image Memorability of Pattern Recognition Machines

## Abstract

We study the problem of measuring and predicting how memorable an image is to pattern recognition machines, as a path to explore machine intelligence. Firstly, we propose a self-supervised machine memory quantification pipeline, dubbed 'MachineMem measurer', to collect machine memorability scores of images. Similar to humans, machines also tend to memorize certain kinds of images, whereas the types of images that machines and humans memorize are different. Through in-depth analysis and comprehensive visualizations, we gradually unveil that 'complex' images are usually more memorable to machines. We further conduct extensive experiments across 11 different machines and 9 pre-training methods to analyze and understand machine memory. This work proposes the concept of machine memorability and opens a new research direction at the interface between machine memory and visual data.

## 1 Introduction

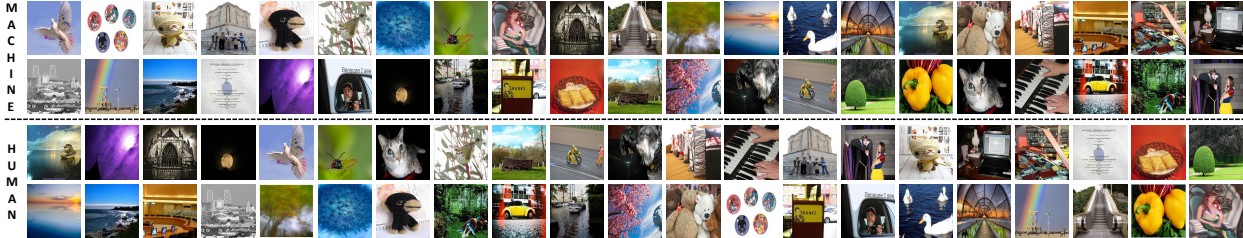

Figure 1: **Sample images** that are sorted from less memorable (left) to more memorable (right) for both machines and humans. The first top two rows are arranged by machine memorability scores and the bottom two rows are sorted by human memorability scores.

In the field of artificial intelligence, pattern recognition machines have been well developed and have become powerful tools, capable of executing complex tasks Radford et al. (2021); OpenAI (2023). However, the goal of achieving human-like intelligence remains challenging, as today's machines often behave very distinctively from humans in many aspects Geirhos et al. (2018); Wichmann & Geirhos (2023).

Memory, a fundamental aspect of both intelligence and cognition, plays a crucial role in both cognition and intelligence Colom et al. (2010). Alongside categorization and reasoning, it forms an integral part of human cognition. As the cornerstone of learning, reasoning, and understanding, memory facilitates the establishment of connections, recognition of patterns, and the making of informed decisions. In an effort to bridge the gap between natural and artificial intelligence, this paper takes a deep dive into the memory of machines. Here, memory of machines refers to the capacity of machines, particularly those parameterized by learned weights, to store and retrieve information. By examining machine intelligence through the lens of memory, we aim to contribute to the development of machines that perceive, learn, and reason in a manner more akin to humans Lake et al. (2017). This approach could potentially bring us a step closer to achieving human-like intelligence in machines.

When humans view images, we naturally perceive similar types of information and tend to behave similarly in memorizing images Khosla et al. (2015). Some images sharing certain patterns are more memorable to us Isola et al. (2013); Khosla et al. (2015); Goetschalckx et al. (2019). This leads us to question: How well do machines memorize certain types of images? What attributes make an image memorable to a machine? Do different machines exhibit varying memory characteristics? In this context, we define machines as models or systems that are capable of learning from data, following the definition used in machine learning, emphasizing the learning and adaptive capabilities of these models or systems. We begin with a quantification of machine memory. We adopt the human memorability score (HumanMem score) Isola et al. (2011) concept and propose the machine memorability score (henceforth referred to as MachineMem score) as a measure of machine memory. Our next goal is to collect MachineMem scores for a variety of images. To accomplish this, we introduce a novel framework, the MachineMem measurer, inspired by the repeat detection task and visual memory game Isola et al. (2013). This framework produces MachineMem scores for images in a self-supervised manner, allowing us to label the entire LaMem dataset Khosla et al. (2015). Based on this, we train a regression model to predict MachineMem scores in real-time and introduce some advanced training techniques to enhance the performance of both human and machine memorability score predictors.

Using collected MachineMem scores, we delve into the investigation of what makes an image memorable to machines. This exploration involves multiple approaches, beginning with a visual teaser (Figure 1), wherein sample images are arranged by their MachineMem scores. A quantitative analysis follows, presenting the correlations between MachineMem scores and 13 image attributes. We then analyze the memorability of different object classes for machines. Further, we apply GANalyze Goetschalckx et al. (2019) to visualize how the memorability of a particular image changes with varying MachineMem scores. We also present a comparative analysis between machine memory and human memory to highlights the differences and similarities. Lastly, we aim to understand machine memorability more deeply. We conduct two case studies that analyze the MachineMem scores produced by 11 different machines and 9 varying pretext tasks.

Our contributions can be summarized as follows:

- We introduce the concept of machine memorability and propose a quantified measure, the Machine-Mem score.

- We develop a MachineMem measurer to collect and quantify MachineMem scores and a MachineMem predictor to predict MachineMem scores in real-time.

- We conduct a comprehensive analysis to uncover and understand the emerging patterns and properties of MachineMem scores.

## 2 Related Work

**Visual cognition and memory.** Pioneering studies Isola et al. (2011; 2013) have systematically explored the elements that make a generic image memorable to humans. They established a visual memory game, a repeat detection task that runs through a long stream of images. This game involves multiple participants, and the averaged accuracy of detecting repeated images provides a quantified HumanMem score for each image. Subsequent research in this area Lahrache & El Ouazzani (2022); Zhang et al. (2020); Bylinskii et al. (2022) has created more datasets Khosla et al. (2015); Bainbridge et al. (2013); Lu et al. (2020); Goetschalckx & Wagemans (2019) and developed more powerful methods for predicting HumanMem scores Kim et al. (2013); Peng et al. (2015); Fajtl et al. (2018); Perera et al. (2019); Lu et al. (2020); Leyva & Sanchez (2021). One of the goals of this work is to compare machine memory and human memory. To this end, we propose the definition of MachineMem score, mirroring the HumanMem score, and incorporate the key design elements of the visual memory game into our MachineMem measurer.

In psychology and cognition research, memory is broadly divided into sensory Pearson & Brascamp (2008), short-term Cowan (2001), and long-term categories Mandler & Ritchey (1977); Vogt & Magnussen (2007); Brady et al. (2008). The visual memory game primarily captures long-term memory Isola et al. (2013). Yet, given that HumanMem scores remain stable over various time delays Isola et al. (2013); Khosla et al.

(2015), they are likely indicative of both short-term and long-term memory Borkin et al. (2015); Cowan (2008). Similarly, our MachineMem measurer, which collects MachineMem scores, takes into account both the short-term and long-term memory capabilities of machines.

**What images are more memorable to humans?** Here, we briefly summarize the characteristics typically associated with human-memorable images:

• Images with large, iconic objects, usually in square or circular shapes and centered within the frame, tend to be more memorable. This suggests that a single iconic object makes an image more memorable than multiple objects.

• Images featuring human-related objects (such as persons, faces, body) and indoor scenes (like seats, floors, walls) have higher HumanMem scores, while outdoor scenes (such as trees, buildings, mountains) generally contribute negatively.

• Bright, colorful images, especially those with contrasting colors or a red hue, are more memorable to humans.

• Simplicity in images often enhances memorability.

Conversely, images that deviate from these trends are generally less memorable Isola et al. (2013); Khosla et al. (2015); Goetschalckx et al. (2019). Furthermore, changes in other cognitive image properties (like aesthetics, interestingness, and emotional valence) show only weak correlations with HumanMem scores. As humans tend to construct simplified representations of the visual world for planning Ho et al. (2022), this may explain why simpler images are typically more memorable.

**Memory modules.** Both implicit Mandler & Ritchey (1977); Chung et al. (2014) and explicit memory Vaswani et al. (2017); Bahdanau et al. (2014); Devlin et al. (2018); Graves et al. (2014); Kumar et al. (2016); Sukhbaatar et al. (2015); Wang et al. (2018); Han et al. (2020) mechanisms have been widely used in designing artificial neural networks to process sequential data. The aim of this paper is not to design new memory modules or enhance the memory capabilities of neural networks. Instead, it focuses on measuring and understanding the MachineMem of visual data.

**Memorization in DNNs.** Previous research Feldman (2020); Zhang et al. (2021); Feldman & Zhang (2020); Toneva et al. (2018); Arpit et al. (2017) has explored the relationship between network memorization and aspects such as capacity, generalization, and robustness in supervised classification tasks. They show more memorable samples are usually easy samples, while more forgettable samples tend to be hard. This is due to DNNs first learning patterns shared by common samples Kishida & Nakayama (2019); Toneva et al. (2018); Feldman & Zhang (2020). In a classification task, the memorization of the network and data are heavily influenced by labels and data distributions. In contrast, our study focuses on the machine memorability of visual data in a more general context, specifically without the use of class labels and supervised classification tasks. This approach allows us to eliminate the effects of labels and data distributions.

## 3 Preliminary: Visual memory game

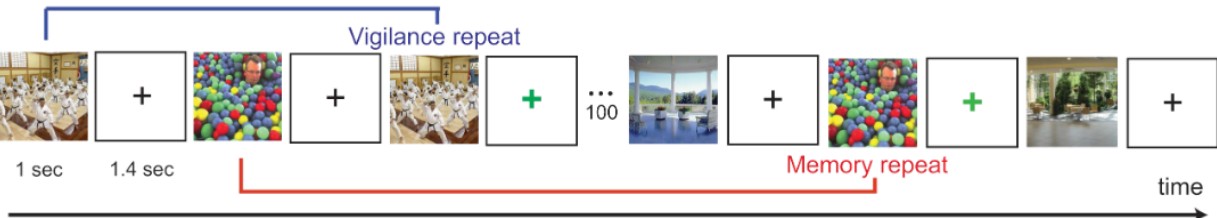

Figure 2: **An overview of the visual memory game.** Participants engage in this game by identifying repeated images in a lengthy sequence. Image sourced from Isola et al. (2011).

The visual memory game is a tool designed to explore the intricacies of image memorability and the processes involved in human memory. As depicted in Figure 2, the game involves presenting participants with a series of images, with the task being to identify any images that are repeated within the sequence. In other words, participants need to determine whether an image has been seen or not. The game allows for the measurement of the likelihood of recalling specific images after a single exposure, thereby providing valuable insights into the factors that influence image memorability.

The game involves multiple participants and the human memorability score of images is calculated as the probability of accurately identifying a repeated image after a single viewing within a lengthy sequence in the game.

## 4 Measure Machine Memory

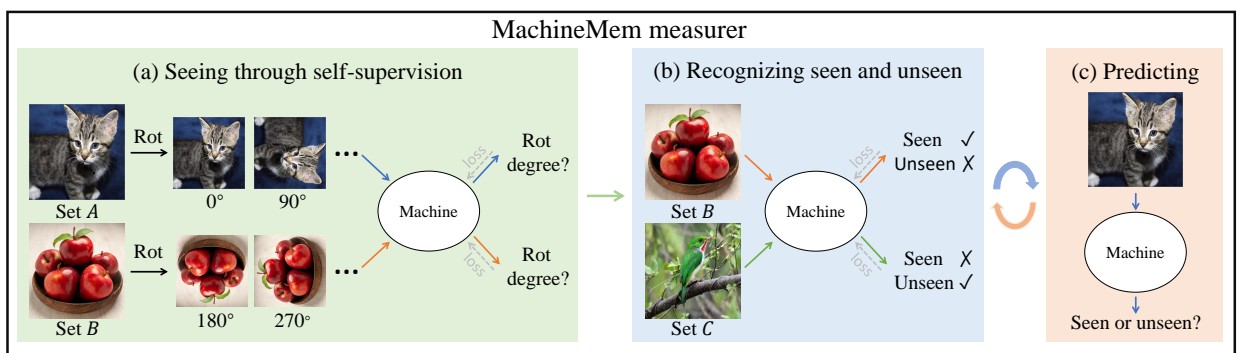

Figure 3: **A realization of the MachineMem measurer**. It consists of 3 stages: (a) Seeing images through self-supervision, (b) Recognizing seen and unseen images, and (c) Predicting whether an image has been seen .Each image we present (cat, apple, and bird) symbolizes an image set ($A$, $B$, and $C$), each containing $n$ images. We focus on measuring MachineMem scores for set $A$ produced by an identical machine. In every episode of the MachineMem measurer, we randomly select sets $B$ and $C$ from an expansive dataset while keeping the cat set constant. The MachineMem scores of set $A$ are obtained by repeating the MachineMem measurer $m$ times.

In this section, our aim is to first provide a formal definition of the machine memorability score (MachineMem score). Following that, we will present our MachineMem measurer, which can collect MachineMem scores in a self-supervised manner.

### 4.1 Define Machine Memorability

Human memorability is defined as *the average percentage of correct detections by multiple humans participants in a visual memory game* Isola et al. (2011). Building upon on this definition, machine memorability is defined as *the average percentage of correct detections by machines during multiple episodes of the MachineMem measurer*. In contrast to humans, machines exhibit diverse structures and memory characteristics. We thus assess multiple machines, reporting individual machine memorability scores. This also provides results of memory characteristics exhibited by each machine. We use a ResNet-50 as our default machine, given its widespread use and strong generalization in various visual tasks.

### 4.2 MachineMem Measurer

We propose the MachineMem measurer that incorporates the same pipeline and fundamental concepts of visual memory game as a pipeline to measure MachineMem scores of images. The visual memory game

presents three integral actions: observe, repeat, and detect. Similarly, we structure the MachineMem measurer as a three-stage process, where each stage corresponds to one of these actions. A conceptual diagram of the MachineMem measurer is presented in Figure 3.

While humans are capable of observing and memorizing images without any feedback mechanism, machines have traditionally lacked this ability. Therefore, in stage (a), we adopt a self-supervision pretext task to guide machines to observe images. Following this, stage (b) instructs the machines to distinguish between observed and unobserved images, thereby equipping the machines to execute the repeat detection task in stage (c). We use LaMem Khosla et al. (2015) as our dataset.

**Seeing through self-supervision.** In the endeavor to help machines observe and memorize images, supervision is indispensable. However, the supervision signal should be self-sufficient. In light of this, we contemplate employing a pretext task as supervision that satisfies three criteria: (1) it necessitates minimal structural modifications at the machine level, (2) it does not degrade or distort the input data, and (3) it allows machines to observe whole images rather than cropped segments. The rotation prediction self-supervision task Gidaris et al. (2018) fulfills all these prerequisites and is therefore selected as the pretext task for stage (a). Other candidate pretext tasks such as masked image modeling may also be considered, but they fail to meet all the criteria, especially (3).

Following common practice Gidaris et al. (2018); Deng et al. (2021), we define a set of rotation transformation functions $G = \{R_r(\boldsymbol{x})\}$, where $R_r$ is a rotation transformation function and $r$ are the rotation degrees $r \in \{0°, 90°, 180°, 270°\}$ . Rotation prediction is a multi-class (4 class here) classification task, where the goal is to predict which rotation degree has been applied to an input image $\boldsymbol{x}$. The loss function is formulated as:

$$\mathcal{L}_{\text{rot}} = \frac{1}{4} \left[ \sum_{r \in \{0°, 90°, 180°, 270°\}} \mathcal{L}_{\text{CE}}(\boldsymbol{y}_r, \boldsymbol{\theta}_m(R_r(\boldsymbol{x}))) \right], \tag{1}$$

where $\boldsymbol{y}_r$ is the one-hot label of $r \in \{0°, 90°, 180°, 270°\}$. $\mathcal{L}_{\text{CE}}$ denotes cross-entropy loss. $\boldsymbol{m}$ is a machine parameterized by $\boldsymbol{\theta}_m$.

Stage (a) uses two sets (sets $A$ and $B$) of images, where images in both two sets are labeled as seen. Half of them (set $B$) go to stage (b) and the other half (set $A$) go to stage (c), as shown in Figure 3. We ensure machines to achieve good performance (top-1 accuracy $\geq 80$ %) on the rotation prediction task. By default, machines without pre-training are trained for 60 epochs in this stage.

**Recognizing seen and unseen images.** This stage aims to teach machines to recognize seen and unseen images. We use a set of seen images (set $B$) that has been used in stage (a) and sample a set of unseen images (set $C$) from a large-scale dataset. A binary classification task targeted at recognizing seen and unseen images is employed. The backbone of the machine remains identical but we replace the 4-way classification layer with a new 2-way linear classification head. The loss function is expressed as:

$$\mathcal{L}_{\text{seen}} = \frac{1}{2} \left[ \sum_{l \in \{\text{seen}, \text{unseen}\}} \mathcal{L}_{\text{CE}}(\boldsymbol{y}_l, \boldsymbol{\theta}_m(\boldsymbol{x})) \right], \tag{2}$$

where $\boldsymbol{y}_l$ denotes the one-hot label of $l \in \{\text{seen}, \text{unseen}\}$, CE is cross-entropy loss, and $\boldsymbol{m}$ stands for a machine parameterized by $\boldsymbol{\theta}_m$.

In our default setting, stage (b) lasts for 10 epochs, and the machine will enter stage (c) upon finishing each epoch.

**Predicting whether an image has been seen.** During the final stage (c), we utilize a set of previously 'seen' images (set $A$) that were not involved in stage (b) for measurement purposes. Here, the machine's task is to discern whether a given image has been seen before, thereby replicating the repeat detection tasks we use to evaluate human memory capabilities.

The process of alternating between stage (b) and stage (c) is performed repeatedly, enabling us to extract multiple measurements from the images in set $A$. This is similar to the validation process in a typical

machine learning training procedure, where we collect multiple validation results but only use the one with the best result for testing. The final outcome of a single MachineMem measurer episode is determined by the iteration that results in the smallest calibration error, which is metric is used to measure the reliability of a probabilistic model's predictions Guo et al. (2017). This selection criterion is based on the premise that a lower calibration error signifies a more precise and dependable measurement. Furthermore, this alternating between stage (b) and stage (c) methodology effectively assesses both the short-term and long-term memory capabilities of the machines, as the iterative nature of stages (b) and (c) may collect final results from both short and long epochs in (b).

**Obtaining MachineMem scores.** Drawing from the HumanMem scores approach Isola et al. (2011), we define the MachineMem score of an image as the ratio of the number of seen predictions to the total number of MachineMem episodes.

During each MachineMem measurer episode, besides set $A$ which is destined for stage (c) for score computation, we randomly select two other sets of images (set $B$ and set $C$) from a dataset, each containing $n$ images. This randomized selection process is designed to ensure that the MachineMem measurer accurately reflects the machine's memory capabilities rather than fitting specific distributions. We default $n$ to 500. To calculate the MachineMem scores for set $A$, we repeat the MachineMem process $m$ times, where $m$ is set to 100, mirroring the average number of participants involved in HumanMem score collection. The MachineMem scores for set $A$ are thus obtained after repeating the MachineMem measurer $m$ times.

We have collected and labeled MachineMem scores for all images in the LaMem dataset (totaling 58,741 images). The average MachineMem score is 0.680 (SD 0.070, min 0.39, max 0.91), which contrasts with the average HumanMem score of 0.756 (SD 0.123, min 0.20, max 1.0).

## 5 Predict MachineMem Scores

Collecting MachineMem scores with the MachineMem measurer can be a time-consuming process, often requiring several hours to generate scores for 500 images. In response to this challenge, we aspire to train a robust regression model capable of predicting MachineMem scores in real-time. This task aligns with the prediction of HumanMem scores, prompting us to revisit and enhance approaches tailored towards predicting HumanMem scores.

Past research Fajtl et al. (2018); Perera et al. (2019) has demonstrated that a modified ResNet-50 regression model (with adjustments to the final layer to accommodate regression tasks) can deliver satisfactory performance in predicting HumanMem scores. This model is trained utilizing dropout Srivastava et al. (2014) and RandomResizedCrop Szegedy et al. (2015). With this training setup, complemented by an ImageNet-supervised pre-training initialization, this simple ResNet-50 regression model can attain a Spearman's correlation, $\rho$, of 0.63 when predicting HumanMem scores. For comparison, the human consistency Khosla et al. (2015) registers at $\rho = 0.68$, while the state-of-the-art result Perera et al. (2019) reaches $\rho = 0.67$.

We show a superior performance can be accomplished in predicting HumanMem scores by employing self-supervised pre-training and strong data augmentations. Specifically, we transfer the knowledge from the pre-trained MoCo v2 Chen et al. (2020b). At the data level, we substitute RandomResizedCrop Szegedy et al. (2015) with CropMix Han et al. (2022b), while integrating Random erasing Zhong et al. (2020) and Horizontal flip applied in a YOCO manner Han et al. (2022a). This results in a ResNet-50 regressor that attains $\rho = 0.69$ in predicting HumanMem scores, surpassing even human consistency! We refer to this model as the enhanced ResNet-50 regression model.

In the subsequent step, we aspire to train a regression model capable of predicting MachineMem scores that align as closely as possible with those derived from empirical observations of 100 trials from the Machine-Mem measurer. We employ and train this enhanced ResNet-50 regression model for the task of predicting MachineMem scores. We randomly select 1000 images as the test set, using all remaining images as the training set. This model also achieves a $\rho = 0.69$ in predicting MachineMem scores. We designate our model as the MachineMem predictor.

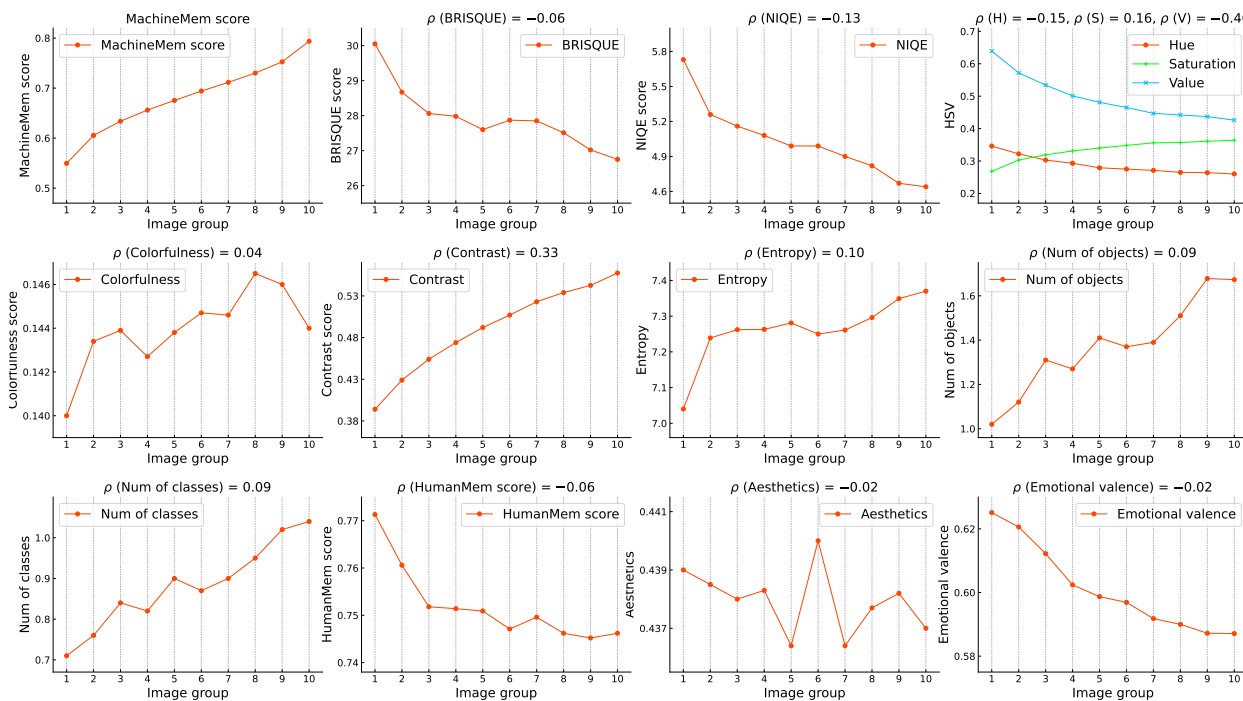

Figure 4: **Relation between image groups and varying image attributes**. We find value and contrast to be the two most significant attributes that correlate moderately ($\rho \geq 0.3$) with MachineMem scores. On the other hand, attributes such as hue and saturation show a weak correlation ($0.15 \leq \rho < 0.3$) with MachineMem scores. Other factors, such as NIQE, entropy, and number of objects, demonstrate a very weak correlation ($0.08 \leq \rho < 0.15$) with MachineMem scores. These findings are based on Spearman's correlation ($\rho$) computed from the entire data set.

## 6 What Makes an Image Memorable to Machines?

This section aims to analyze MachineMem scores in order to understand what factors contribute to an image's memorability for machines. We present some sample images in Figure 1, first analyzing the relationship between MachineMem scores and 13 image attributes. With the aid of the MachineMem predictor, we predict MachineMem scores of all ImageNet Russakovsky et al. (2015) training images to analyze which classes are most and least memorable to machines. Additionally, we employ the GANalyze Goetschalckx et al. (2019), capable of adjusting an image to generate more or less memorable versions, as a means to discover hidden trends that could potentially influence MachineMem scores. In conjunction with GANalyze, we conduct a comparative study between machine memorability and human memorability.

### 6.1 Quantitative analysis

Do image attributes adequately determine MachineMem scores? Here we examine 13 image attributes, roughly grouped into 4 categories, each focusing on different measurements. Based on MachineMem scores of images, we sort all LaMem images and organize them into 10 groups, from the group with the lowest mean MachineMem scores to the highest. Each group contains approximately 5870 images. Spearman's correlation results are computed based on all 58741 images. Figure 4 presents plots illustrating the relationship between image groups and varying image attributes.

**Image quality.** We employ two metrics, NIQE Mittal et al. (2012b) and BRISQUE Mittal et al. (2012a), to assess image quality. Lower NIQE and BRISQUE values suggest better perceptual quality, indicating fewer distortions.

Both BRISQUE and NIQE demonstrate a very weak correlation with MachineMem scores ($\rho = -0.06$ and $-0.13$, respectively). NIQE shows a relatively stronger correlation, possibly because BRISQUE, which involves human subjective measurements, aligns more with human perception.

**Pixel Statistics.** We investigate the correlation between MachineMem scores and basic pixel statistics. Hue, saturation, and value from the HSV color space Agoston (2005) are measured, along with colorfulness Hasler & Suesstrunk (2003), contrast Matkovic et al. (2005), and entropy.

Interestingly, value and contrast show substantial correlations with MachineMem scores ($\rho = -0.40$ and $-0.33$, respectively). Deep color and strong contrast are two significant factors that make an image memorable to machines. Hue and saturation are weakly correlated with MachineMem scores ($\rho = -0.15$ and $0.16$, respectively). Entropy exhibits a very weak correlation with MachineMem scores ($\rho = 0.10$). However, as presented in Figure 4, the group with the lowest MachineMem scores, *i.e.*, group 1, displays very low entropy. Images with very low MachineMem scores often lack contrast or have a light color background (see Figure 1), and therefore tend to have low entropy. Furthermore, colorfulness seems to have no clear correlation with MachineMem scores ($\rho = 0.04$), except for the fact that group 1 scores very low in terms of colorfulness.

**Object Statistics.** We measure the number of objects and the number of classes (unique objects) within an image. A YOLOv4 Bochkovskiy et al. (2020) model is employed as the object detector.

Both metrics are very weakly correlated with MachineMem scores (same $\rho = 0.09$). By excluding data with 0 objects, their correlations with MachineMem scores are still very weak (same $\rho = 0.10$).

**Cognitive Image Property.** We employ HumanMem scores, aesthetics, and emotional valence as cognitive image properties. The HumanMem scores are obtained from the LaMem dataset. To measure aesthetics, we utilize a pre-trained LIMA model Talebi & Milanfar (2018). We use the emotional valence predictor from the GANalyze for determining emotional valence.

The correlation between HumanMem scores and MachineMem scores is very weak ($\rho = -0.06$). Similarly, other cognitive image properties, such as aesthetics and emotional valence, exhibit negligible correlation with MachineMem scores (both with $\rho = -0.02$). These findings suggest that MachineMem scores represent a unique image property that is distinct from other image properties.

In conclusion, unlike human memory, which is largely driven by semantics, machines, devoid of such common sense, tend to emphasize more on basic pixel statistics.

## 6.2 What classes are more or less memorable?

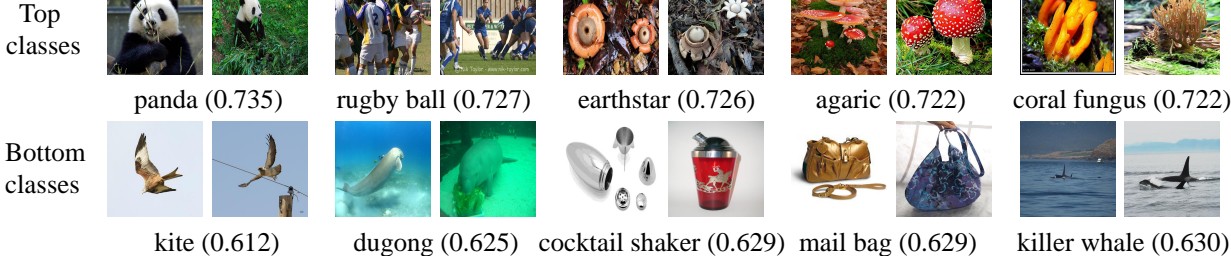

Figure 5: **ImageNet objects sorted by their mean MachineMem scores**. We report the top-5 and bot-5 classes and their mean MachineMem scores. It appears that the classes ranking highest commonly exhibit lower values paired with pronounced contrast. To illustrate, images of pandas consistently feature a mix of both white and black hues, often juxtaposed against a green background, thus enhancing the overall contrast. Conversely, classes ranked at the bottom predominantly showcase lighter backgrounds occupying a substantial proportion of the pixel space.

Do images belonging to certain classes tend to be more or less memorable to machines? We use the Machine-Mem predictor to predict MachineMem scores of all ImageNet training images to obtain mean MachineMem

scores of 1000 ImageNet classes. Figure 5 summarizes the top and bottom classes. By analyzing gained results, we find the answer to be yes: Classes containing light backgrounds are usually less memorable to machines, for instance, classes related to sea or sky. Classes that have strong contrast, high value, and multiple objects tend to have high MachineMem scores. To further substantiate this observation, we examined the standard deviation within each class. For the top classes, the standard deviations are relatively small, mostly ranging from 0.024 to 0.033. Conversely, for the bottom classes, the standard deviations can be higher, varying from 0.03 to 0.057. For context, the average standard deviation across all classes is 0.046, which is relatively small.

## 6.3   How do images change with varying MachineMem scores?

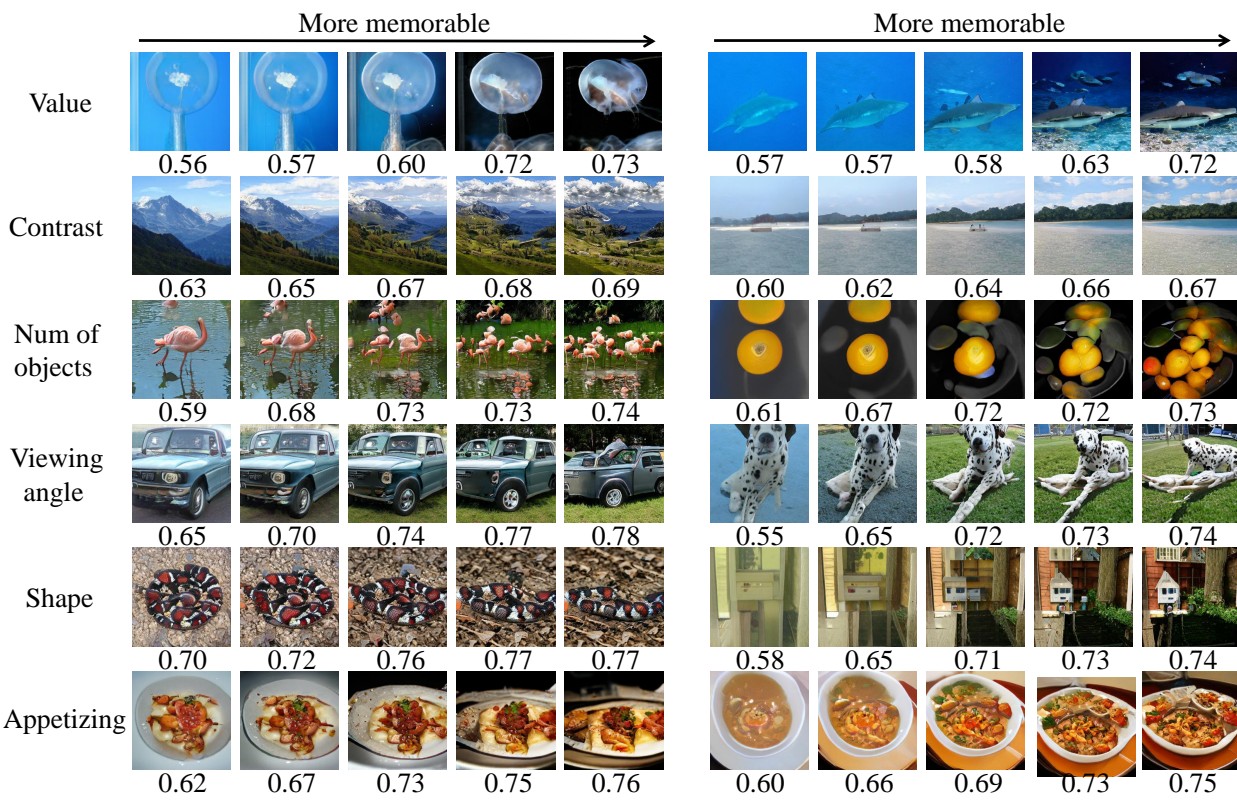

Figure 6: **Images generated by GANalyze**. We visualize what will happen if we make an image more or less memorable to machines. We summarize 6 trends, where the first 3 of them (value, contrast, and number of objects) are previously found in the quantitative analysis part. We show the results of GANalyze as further confirmations. For certain objects, viewing angle, shape, and appetizing are hidden trends that are unveiled by GANalyze. An overall trend is that GANalyze is often complexifying images to make them more memorable to machines.

While the relationship between MachineMem scores and various image attributes has been established, certain concealed factors that potentially change an image's MachineMem remain elusive. Therefore, we leverage the potent capabilities of GANalyze to uncover these hidden elements that could influence MachineMem scores. More specifically, we employ the MachineMem predictor as the Assessor within GANalyze to guide the model in manipulating the latent space to change an image's machine memorability. We manipulate the latent space using a relatively large step size, alpha (0.2). This could potentially amplify trends that influence MachineMem scores. The results of this investigation are depicted in Figure 6.

In the process, we also utilize GANalyze to provide additional validation for the correlations between MachineMem scores and image attributes. We elucidate three observable trends: Value, Contrast, and Number

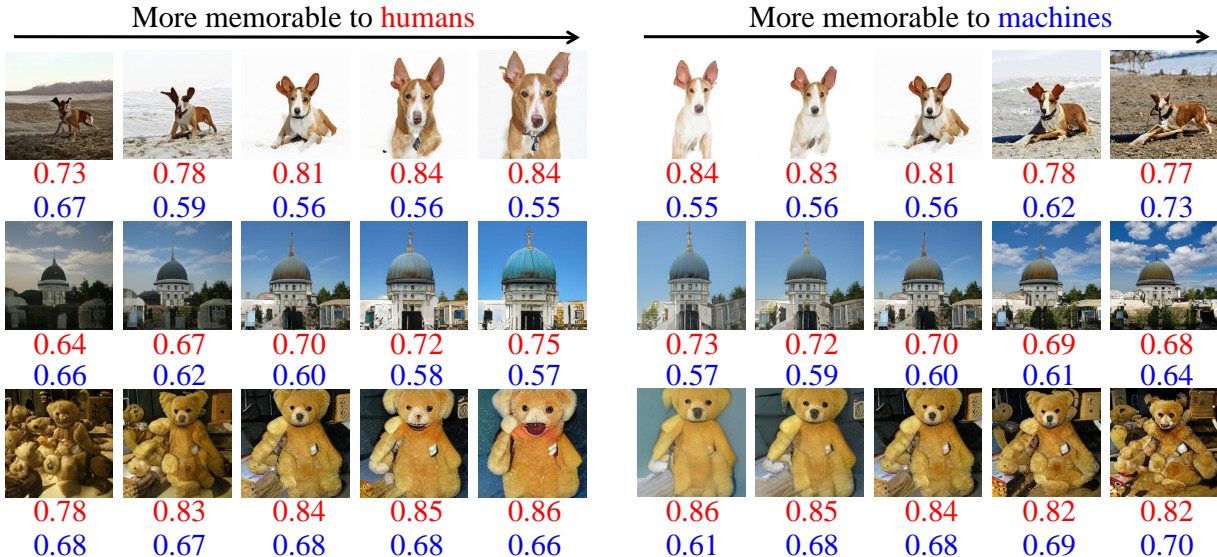

Figure 7: **A comparison between human memory and machine memory**. We employ GANalyze to make an image more as well as less memorable to both humans and machines. HumanMem scores are labeled in red while blue indicates MahinceMem scores. Generally speaking, simple images are more memorable to humans while complex images are more memorable to machines.

of Objects. In terms of concealed trends, we identify three standout candidates. The newly unearthed trends are:

• Viewing angles that provide more information (here side viewed) are usually more memorable than those angles providing less information.

• Objects in standard shapes (square or circular) seem to be less memorable to machines, whereas objects in irregular shapes tend to be more memorable to machines.

• For food-related objects, images with high MahcineMem scores often look appetizing.

A critical overarching trend observed across almost all images is the machine tend to memorize complex images. Factors such as contrast, number of objects, viewing angle, shape, and appeal to taste can all be considered manifestations of this pervasive trend.

## 6.4 Human memory vs. machine memory

As presented in Figure 4, MachineMem scores and HumanMem scores are very weakly correlated ($\rho = -0.06$). But in GANalyze, which is good at showing global trends, we find machines tend to memorize more complex images, which is on the reverse side of humans that are usually better at memorializing simple images. Such results are presented in Figure 7. Other than ResNet-50, we also explore the correlations between multiple machines (10 other different machines that will be presented in the appendix) 12 and humans, however, none of these machines show clear correlations ($|\rho| \geq 0.15$) with humans. This further suggests MachineMem is very distinct from HumanMem.

## 7 Understanding Machine Memorability

HumanMem, as an inherent and consistent attribute of images, is universally recognized by individuals, transcending their diverse backgrounds Isola et al. (2013). This implies that, despite varying human experiences, there exists a shared element in how humans remember visual data. But do machines exhibit a similar

principle? Here we delve into two key questions: Will MachineMem scores remain consistent across different machines? What role does varying pre-training knowledge play?

## 7.1 Memorability across machines

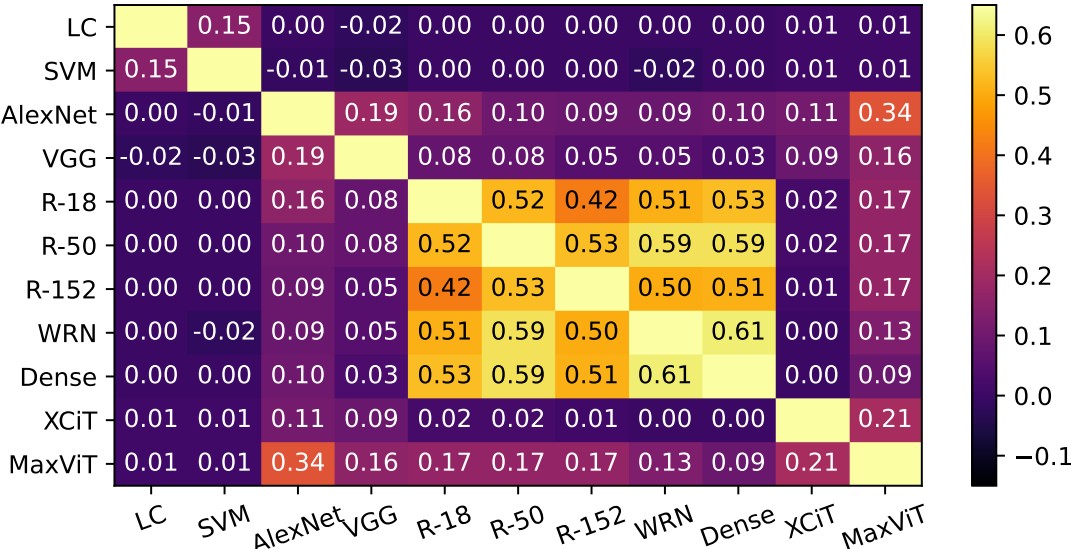

Figure 8: **Memorability across machines**. Each off-diagonal corresponds to Spearman's correlation ($\rho$) of two machines. Machines within each category are usually strongly correlated, but this trend does not scale to machines across categories.

We scrutinize the relationships among 11 distinct machines, grouping them into four categories: conventional machines (linear classifier and SVM Cowan (2001)), classic CNNs (AlexNet Krizhevsky et al. (2012), VGG Simonyan & Zisserman (2014)), residual CNNs (ResNet-18, ResNet-50 He et al. (2016), ResNet-152, WRN-50-2 Zagoruyko & Komodakis (2016), and DenseNet121 Huang et al. (2019)), and ViTs (XCiT-T Ali et al. (2021) and MaxViT-T Tu et al. (2022)). We examine and evaluate the MachineMem scores of 10000 LaMem images as produced by these varying machines. Due to the inherent constraints of conventional machines, we employ a binary classification task (0 °and 90 °comprising one class, and 180 °and 270 °forming the other) as their pretext tasks in the initial stage (a) of the MachineMem measurement process. The training parameters are identical for each machine within the same category, although slight variations exist across different categories.

As illustrated in Figure 8, machines within the same category often exhibit strong correlations. Over 50% of instances show this trend, even when considering the outlier case of AlexNet and MaxViT. If we disregard this outlier, the percentage rises to 100%. Notably, residual CNNs display the strongest correlation, with an average $\rho$ value of 0.53. This suggests a strong propensity for these machines to memorize similar images. However, less apparent correlations are observed among machines from different categories. For example, conventional machines do not correlate well with machines from other categories, likely due to their limited capabilities.

## 7.2 Memorability across pre-training methods

We explore nine pre-training methods applicable to a ResNet-50 model. This includes supervised ImageNet classification pre-training and eight unsupervised methods, such as relative location Doersch et al. (2015), rotation prediction Gidaris et al. (2018), PIRL Misra & Maaten (2020), DeepCluster-v2 Caron et al. (2021), and four instance discrimination approaches (NPID Wu et al. (2018), SimCLR Chen et al. (2020a), MoCo v2 Chen et al. (2020b), and SimSiam Chen & He (2021)). The analysis is conducted on 10,000 LaMem images.

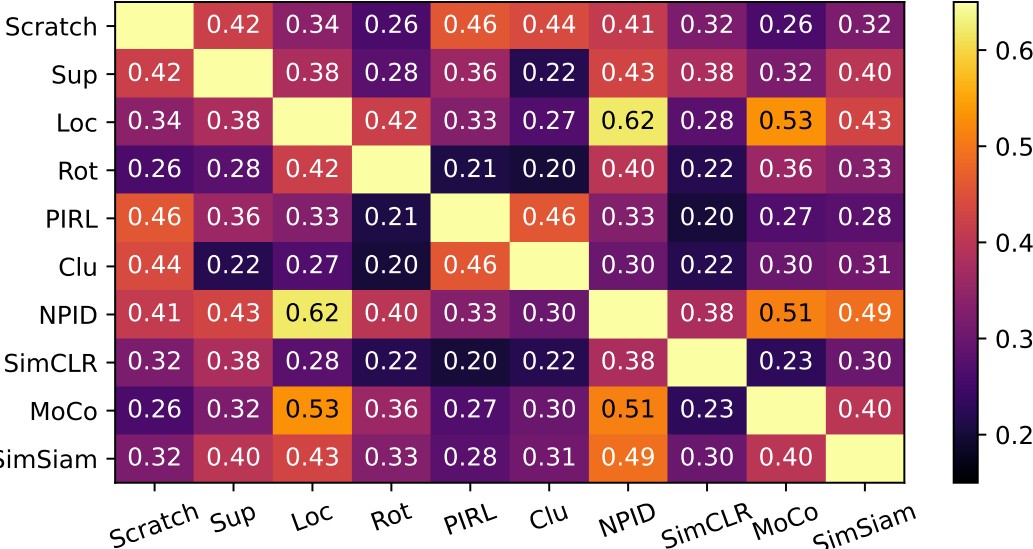

Figure 9: **Memorability across pre-training methods**. Spearman's correlation ($\rho$) of two pre-training methods is presented at each off-diagonal. Though having different prior knowledge, an identical structured machine tends to memorize similar images.

Figure 9 summarizes our findings. The highest correlation in MachineMem scores ($\rho = 0.62$) is observed between location prediction and NPID, while the weakest correlation ($\rho = 0.20$) is found between PIRL and SimCLR. In general, the memory capabilities of a ResNet-50 model are influenced by its pre-training knowledge, yet moderate correlations (average $\rho = 0.35$) are still present, indicating that machines, regardless of pre-training knowledge, tend to memorize similar images in a moderate manner. One explanation could be that, although pre-training tasks are designed differently, they all aim to learn a strong and transferable representation. As a result, they are expected to share some similarities in terms of memory characteristics.

## 8 Discussion

Identifying visually memorable data can lead to practical applications in areas such as data augmentation, continual learning, and generalization. For instance, we could design a new data augmentation strategy that makes data more memorable to machines, thereby aiding the training of neural networks. To explore this, we conduct a preliminary experiment. In this experiment, we measure the machine memorability scores of different augmentation methods applied to the same image set. The MachineMem score without augmentation is 0.678, but with the application of widely used augmentation methods such as AutoAug Cubuk et al. (2018) and ColorJittering, the scores improve to 0.691 and 0.685 respectively. This suggests that data augmentation can indeed impact machine memorability scores. In the context of continual learning, it may be advantageous to prioritize data that is less memorable.

The way artificial intelligence operates is still vastly different from natural intelligence and creating machines that mimic human behavior remains challenging. A deeper comprehension of how pattern recognition machines work can facilitate the development of more intelligent machines.

## 9 Conclusion

We propose and study a property of images, *i.e.*, machine memorability. Machine memorability shows a cognitive property of machines and can serve as a pathway to help us to further explore machine intelligence. We hope our findings could provide insights into fundamental advances in computer vision, machine learning, natural language processing, and general artificial intelligence.

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

## A  Training details of MachineMem measurer

In our research, we investigate the memory characteristics of 11 distinct machines. These machines are categorized into four groups, namely conventional machines (comprising linear classifier and SVM Cowan (2001)), classic CNNs (such as AlexNet Krizhevsky et al. (2012) and VGG Simonyan & Zisserman (2014)), residual CNNs (including ResNet-18, ResNet-50 He et al. (2016), ResNet-152, WRN-50-2 Zagoruyko & Komodakis (2016), and DenseNet121 Huang et al. (2019)), and ViTs (like XCiT-T Ali et al. (2021) and MaxViT-T Tu et al. (2022)).

Except for the number of training epochs in stage (a) and the corresponding learning rate, all training hyperparameters remain consistent across all machine types and pre-training methods. We have made these adjustments to ensure machines are able to achieve satisfactory performance levels (top-1 accuracy $\geq 80\%$) during stage (a).

Our MachineMem measurer is trained solely on a single GPU, with a batch size of 1 to parallel the visual repeat game settings. We employ SGD as our optimization algorithm and use a cosine learning schedule for our training process. The settings for weight decay and momentum are 0.0001 and 0.9, respectively. The specifics for the training epochs in stage (a) and learning rates for all machine models are as follows:

Conventional machines: Training epochs for stage (a) are set at 60, with a learning rate of 0.01.

Classic CNNs: Training epochs for stage (a) are set at 70, with a learning rate of 0.0005.

Residual CNNs: Training epochs for stage (a) are set at 60, with a learning rate of 0.01.

ViTs: Training epochs for stage (a) are set at 70, with a learning rate of 0.0005.

ResNet-50 with pre-training: Training epochs for stage (a) are set at 30, with a learning rate of 0.01.

## B  Training details of MachineMem predictor

MachineMem predictor and HumanMem predictor share identical training settings. We use a MoCo v2 model (800 epochs pre-training) as initialization. The prediction model is trained for 30 epochs with an SGD optimizer. Weight decay and momentum are set as 0.0001 and 0.9, respectively. The batch size is 100. The initial learning rate is 0.01 with a cosine decay schedule. For CropMix, the crop scale is set as (0.8, 1.0). We use CutMix as the mixing operation and color permutation as the intermediate augmentation.

## C  Details and analysis of calibration

In stage (c), we adopt the calibration error metric to enhance the reliability and validity of our results by measuring images in set $A$. Regarding particular methods, RMS calibration error Hendrycks et al. (2018) and adaptive binning Nguyen & O'Connor (2015) are employed.

To improve the robustness of our method, we have further enhanced our original approach to incorporate a held-out set of images for the assessment of calibration quality in both seen and unseen image categories. We

have validated this enhanced strategy across several neural network models and found that the MachineMem scores produced align closely with our original results, which were based solely on seen images. This was supported by a strong Spearman's correlation ($\rho > 0.6$) across all tested machines. These findings substantiate that assessing calibration quality using seen images alone is adequate for reliable measurements.

## D    What scenes are more or less memorable?

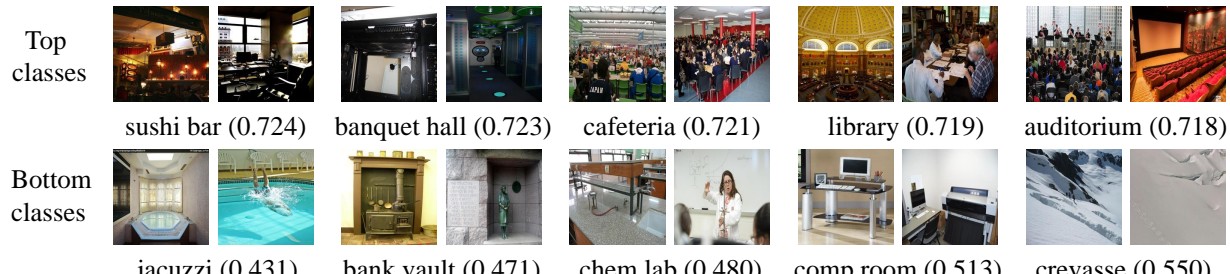

Figure 10: **Places scenes sorted by their mean MachineMem scores**. The top-5 and bot-5 scenes and their mean MahcineMem scores are reported. As observed in the ImageNet classes, the scenes with higher MahcineMem scores generally exhibit lower value, higher contrast, and contain multiple objects. Alternatively, the scene with lower MahcineMem scores often feature white walls and while outdoor scenes.

Do the trends observed from ImageNet classes apply to scenes as well? To answer this, we present the top and bottom Places scenes in Figure 10. We utilized the MachineMem predictor to estimate MachineMem scores for all the training images in Places365 Zhou et al. (2017), thus enabling us to compute average MachineMem scores for 365 scenes. The results indicate that the patterns identified in classes/objects are also evident in scenes, suggesting that this trend is broadly applicable to visual data.

## E    Can more or less memorable classes be semantically grouped according to a hierarchical structure?

| Top-5 | basidiomycete 0.722 | procyonid 0.720 | player 0.716 | marketplace 0.716 | fungus 0.714 |
|---|---|---|---|---|---|
| Bot-5 | rescue equipment 0.606 | computer 0.607 | reservoir 0.608 | sailing vessel 0.612 | hawk 0.612 |

Table 1: ImageNet supercategories sorted by their mean MachineMem scores. We report top-5 and bot-5 supercategory and their mean MahcineMem scores.

We utilized ImageNet's supercategories to delve into this question. Table 1 outlines the top-five and bottom-five ImageNet supercategories, together with their average MachineMem scores. These findings align with our class-level observations, confirming that memorable classes can indeed be semantically grouped according to a hierarchical structure.

## F    Is MachineMem consistent across training settings?

Human memory remains consistent over time Isola et al. (2013). In a visual memory game, an image that is notably memorable after a few intervening images retains its memorability even after thousands of intervening images. We evaluate a ResNet-50 model across different training settings to discern if MachineMem shares this consistency over varying training configurations.

**Number of samples.** By default, we select the image set size $n$ as 500. We further test this setting with sizes of 50, 1000, and 5000. The Spearman's correlation between the default setting $n = 500$ and these

variations are $\rho = 0.05$, $\rho = 0.66$, and $\rho = 0.43$ respectively. A very small $n$ is insufficient to train a stable model and thus fails to accurately reflect memory characteristics. As $n$ increases, the correlations between different $n$ values become increasingly strong.

**Number of epochs in stage (a).** The default number of epochs in stage (a) is set to 60. We test four additional settings: 15, 30, 45, and 75. The Spearman's correlation between the default setting and these variants are $\rho = 0.21$, $\rho = 0.42$, $\rho = 0.55$, and $\rho = 0.57$ respectively. The correlation becomes stronger as the number of epochs in stage (a) increases. Once the model undergoes sufficient training (30 epochs), its memory characteristics stabilizes over multiple epochs in stage (a).

**Number of epochs in stage (b).** We default the number of epochs in stage (b) to 10. Four other settings (1, 4, 7, and 13) are tested. The Spearman's correlation between the default setting and these variants are $\rho = 0.31$, $\rho = 0.54$, $\rho = 0.59$, and $\rho = 0.57$. As with human memory, machine memory also appears consistent over time/delay. This finding suggests that for both humans and machines, the correlation between short-term and long-term memory remains strong in such rank memorability measurements.

## G   Validating Trends that Change MachineMem Scores

In section 5.3 of the main paper, 3 newly discovered hidden trends (Viewing angle, shape, and tasty) are unveiled. However, in the GANalyze framework, semantics are not disentangled, *i.e.*, when transferring an image to more or less memorable versions, many semantics are changing together. In this section, we validate these newly discovered trends.

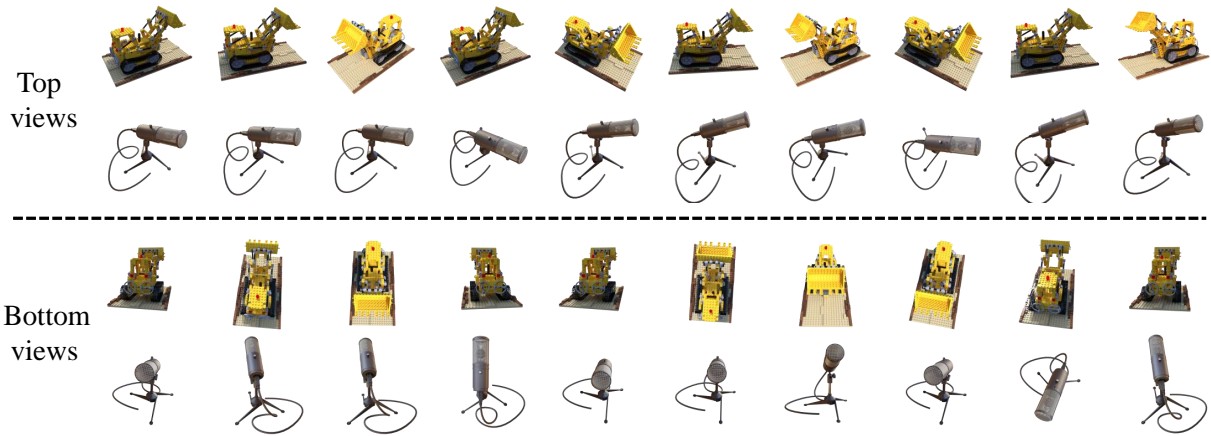

Figure 11: **Scene views sorted by their MachineMem scores**. For each scene, we report top-10 and bot-10 views. Top views are with viewing angles that tend to present more information. For instance, top lego views are usually side viewed while bot lego views are viewed in the front or back.

**Viewing angle.** We study two scenes (lego and mic) from the nerf-synthetic Mildenhall et al. (2021) dataset. For each scene, we show the top-10 and bottom-10 images/views in Figure 11, where the top views usually contain more information (more parts of objects presented) than the bottom views. Results here further validate our finding on viewing angles, *i.e.*, viewing angles that provide more information are usually more memorable to machines.

**Shape.** For both regular and irregular shapes, we draw 25 images as test sets. We transform every image using 4 rotation degrees $\{0°, 90°, 180°, 270°\}$ to extend the number of images to 100. The mean MachineMem score of the regular shapes image set and the irregular image set is identical, 0.61. Though this trend is shown in GANalyze, results here suggest that shape along might not be able to determine MachienMem scores.

**Tasty.** We collect 100 images from the internet with the keyword "tasty food" as an image set of tasty food. Another image set, not tasty food, which also contains 100 images, is collected using two keywords "disgusting food" and "overcooked food". The mean MachineMem score of the tasty image set is 0.70 while the non-tasty image set is 0.66. This further suggests that food-related images with a higher MahcineMem score often look tastier.

## H    What images are more memorable to other machines?

We further incorporate three additional models including AlexNet Krizhevsky et al. (2012), DenseNet121 Huang et al. (2019), and MaxViT-T Tu et al. (2022) to visualize the types of images they find more memorable. As depicted in Figure 12, we employ GANalyze to facilitate a comparative visual representation across multiple machine models. Factors such as value, contrast, viewing angle, appetizing, and complexity continue to serve as reliable predictors of MachineMem scores across various machine models. However, metrics like the number of objects and shape do not consistently apply to all machines. For instance, MaxViT does not display a preference for images containing a larger number of objects. Despite this, the overarching trend remains, that is, machines generally find more complex images memorable. In the following section, we will present a more thorough quantitative analysis concerning machine memorability across various machine models.

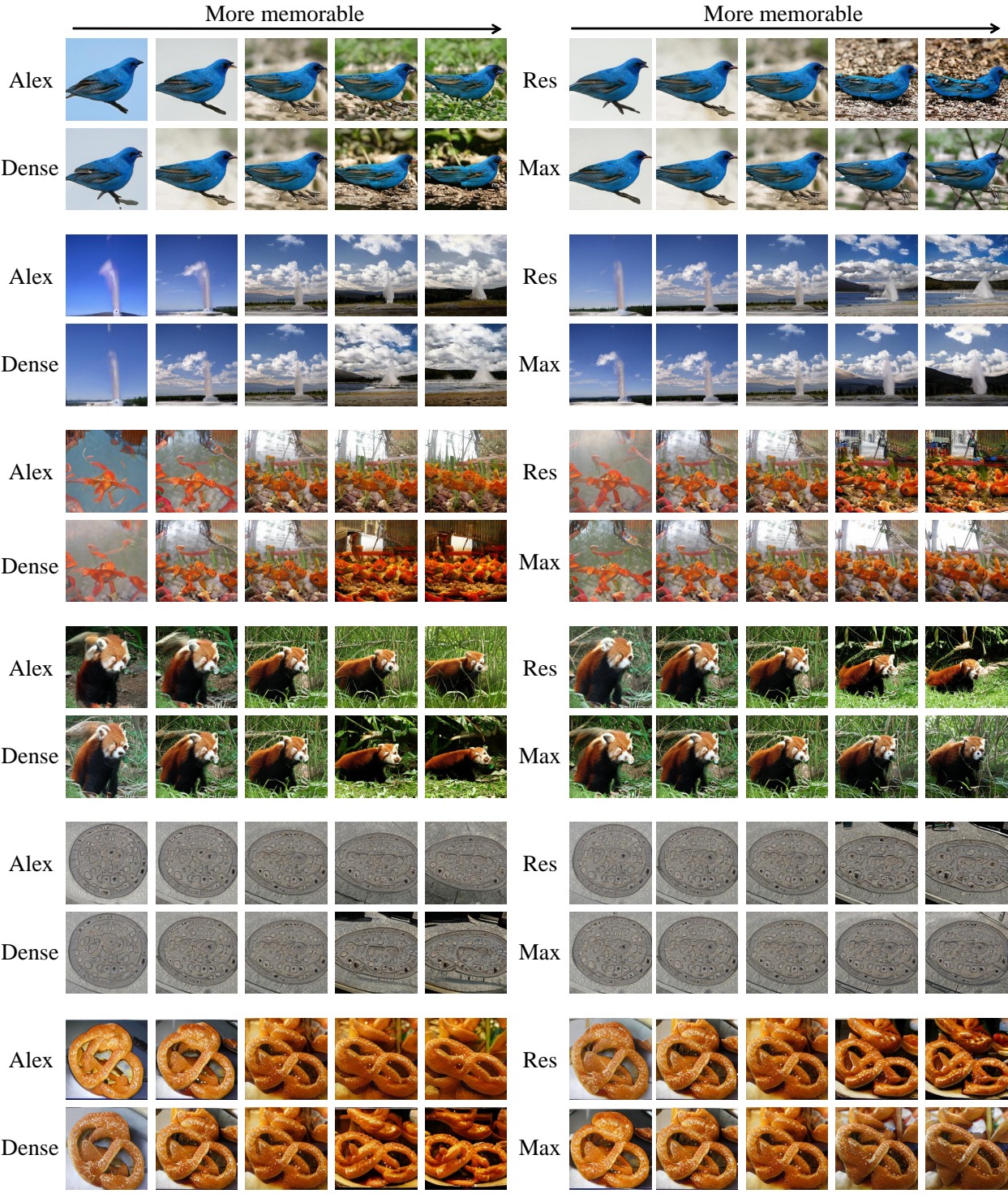

Figure 12: **A comparison between multiple machines using GANalyze**. Alex, Res, Dense, and Max are abbreviations representing AlexNet, ResNet-50, DenseNet, and MaxViT respectively. Each pair of rows represents a distinct trend, in the following order from top to bottom: value, contrast, number of objects, viewing angle, shape, and appetizing. We continue to discern patterns that recur across diverse machine models. For instance, images securing higher MachineMem scores typically exhibit lower value and strong contrast. Most trends identified within the ResNet-50 translate to other machines.

