# OpenReview forum: "On the Notion of Image Memorability of Pattern Recognition Machines"
_TMLR — Rejected by TMLR_

### Review · Reviewer_RWb5 · 2024-04-13

**Summary Of Contributions:**

This paper aims to formulate and investigate machine memorability of images, i.e. a measure of how memorable an image is to artificial neural networks trained on image-related tasks. The primary contributions are:
- MachineMem score - a score inspired by the HumanMem score, quantifying memorability of images to trained ANNs. Specifically, the average percentage of correct predictions of whether or not a (previously seen) image has been seen before during multiple phases of the MachineMem measurer
- MachineMem measurer - a three-phase training/inference scheme for testing memorability of images to the network
- Analysis of what makes image memorable to machines
  - Quantiative analysis of memorability of four categories of image attributes - image quailty, pixel statistics, object statistics, cognitive image properties
  - Comparison of MachineMem and HumanMem scores on various image attributes
  - Analysis across different architectures and pretraining methods

**Audience:**

Yes

**Claims And Evidence:**

No

**Requested Changes:**

- More structured writing. More subsections (and maybe fewer sections - there are a couple that are brief and purely prose), or at least more clearly signposted claims. Better organization in paragraphs, more use of lists, etc. This is crucial to making this paper ready for acceptance.
- Stronger descriptive statistics and specific backing for claims - see weaknesses for places where claims seem unsubstantiated. This is even more crucial for this paper to be acceptance-ready. Right now, most of the analysis is simply not very meaningful.

This is a brief list, but specifics can be found in the weaknesses section - addressing each of those points would help.

**Strengths And Weaknesses:**

## Strengths
- Paper is well-motivated - I can see many benefits to understanding what types of images are more likely to be persistently recognized by NNs even as training continues. The discussion section briefly but clearly outlines the potential of such an analytical method.
- MachineMem score has an interesting and useful design, even more so because it is similar to HumanMem. It's always useful to uncover similarities between perceptual systems, as it sets up interesting future work.
- Experimental structure is solid. The paper investigates many aspects of memorability - attributes that make an image memorable, similarities and differences between human and machine image memorability, testing under various architectures and pretraining methods - that would come up. This approach of addressing a variety of questions is good to see in an analytical paper.
- Prose is clear and descriptive. At each point, I have a clear understanding of the approach so far (aside from a few small things I've flagged below). I felt that I came away from the paper without any major clarification questions
- Figure 7 is very compelling because of the monotonic trends. It is ultimately samples, but it builds intuition and is an interesting result.

## Weaknesses
### Content
- Several of the claims lack strong stats - e.g. the claim about low MachineMem images lacking contrast or having light color background
- Analysis is inconsistent - the fact that entropy is weakly correlated with MachineMem scores but group 1 scores low on entropy is treated as significant. Colorfulness is presented as being the same (weakly correlated with MachineMem, low scores for group 1) but this is treated as insignificant. All of the analysis is in prose, which makes it sound like speculation.
- There isn't much justification for the selection of attributes. They all make sense, but (especially as we get into higher-level attributes) there's no reason to believe they're extensive. Given that they use GANalyze out of the box, it's not proven that high-level concepts aren't memorable to machines, but only that the instances of them that apply to humans are not (if my understanding of GANalyze is correct)
- Stats are weak - a handful of single numbers presented for everything
  - The most notable example is of classes - in table 1, we don't see standard deviation. I suspect it's very high, which is relevant if the paper wants to claim that certain classes have high memorability. Especially because "strong contrast", "high value", and "multiple objects" are fairly divorced from high-level content (a claim the paper rests on earlier), so it's not intuitive (or proven) that certain classes have significantly and robustly higher contrast, value, and object counts across all images than others
  - Furthermore, the paper claims that classes with multiple objects tend to be high, but previously we were told number of objects was only weakly correlated with MachineMem and that it was not a notable factor of memorability
  - Claims about informative viewing angles, irregular shapes, and "appetizing" food (defined how?) rest purely on examples
  - Figure 8 hardly backs up the claims made about it. The idea that "machines within the same category generally exhibit strong correlations" is only true for modern CNNs. Correlation is slightly higher within group for other groups, but it's still weak. There are also violations - for example, MaxViT is more correlated with AlexNet than XCiT
  - Figure 9 is also hard to draw conclusions from. What would I expect if memory capabilities were *not at all* influenced by pretraining knowledge - would I expect all these to be perfectly correlated? For one, it's not clear - I would appreciate a bit more handholding about what exactly it means for scores from these two methods to be correlated. For another, my intuition is that I would expect a fully perfectly correlated graph if memory capabilities were not at all influenced by pretraining knowledge. Given that figure 9 is full of correlations lower than 0.5, I don't see justification for the claim that memory capabilities of ResNet-50 are not significantly influenced by pretraining knowledge.

#### Questions
- From what I understand, stage (c) doesn't involve previously unseen images. Why not? How would we reason about unseen images being predicted as "seen"? Is that not an important aspect of memorability? If I'm wrong about stage (c) not considering previously unseen images, why are those involved in the analysis?
- What makes modern CNNs a distinct category from classic CNNs? The residual objective?
### Presentation
- Overall - writing is verbose and lacks structure. Though it's clear, it can get tiring to read.
- In the intro, it would help to have contributions clearly bulleted and listed instead of just spelled out at a high level in prose
- In the related work, stage (b) of the MachineMem measurer is referred to before it's defined - this is confusing. It should either be described instead of just being referred to, or an explanation should be provided beforehand
- "without the constraints of labels and data distributions" - what exactly is meant by saying this paper doesn't use data distributions?
- Explanation of MachineMem in section 3.1 is unclear. I realize that it's introductory, and the questions that come up are answered soon afterward, but it would be better to just skip it entirely than to have writing that is unclear. The writing feels less like it's signposting upcoming material, and more like it's offering an incomplete explanation, which is frustrating to read and not necessary because the rest of the section is good. Questions I wrote down when reading it that were answered later:
  - Is something changing iteration by iteration? I assume training continues, but it's not clear from the prose.
  - What do you mean by "multiple episodes"? What constitutes an episode?
- Explain the calibration error metric briefly. "Higher degree of reliability and result accuracy" sound like benefits you could get from basic accuracy, so it would help to explain what exactly calibration error is and, more importantly, why you chose it here.

---

> ### Author Response · Authors · 2024-05-06
> **Thank you for your comments**
>
> We're grateful for your valuable comments on our manuscript. Your input has significantly improved our work.
>
> Q1: Several of the claims lack strong stats - e.g. the claim about low MachineMem images lacking contrast or having light color background.
>
> We found such trends mostly from categories/classes. Classes that typically contain light backgrounds or with low contrast tend to have low memorability scores. Indeed, it could be better to provide some statistical numbers to support such claims. To do this, we trained a classifier on light background, dark background, and others (with searching and retrieval to collect data, around ~1000 images per class). To this end, we classified 10000 ImageNet images, and the class with a light background has a mean machine memorability score of 0.62, while the class with a dark background has a mean of 0.70. This could provide more evidence supporting our claim.
>
>
> Q2: Analysis is inconsistent - the fact that entropy is weakly correlated with MachineMem scores but group 1 scores low on entropy is treated as significant. Colorfulness is presented as being the same (weakly correlated with MachineMem, low scores for group 1) but this is treated as insignificant.
>
> Entropy and colorfulness (with Spearman correlations of 0.10 and 0.04) do share similar trends, like they are not strongly correlated with MachineMem scores except for group 1. Overall the trend in terms of correlation for entropy is generally stronger than colorfulness, this is also reflected in Spearman correlation, so we draw a slightly different conclusion on these two metrics. Figure 4 in the revised manuscript also presents our standards for stating correlation levels (where Spearman correlation ranges of 0- 0.08 and 0.08 - 0.15 are for no correlation and very weak correlation).
>
>
>
> Q3: There isn't much justification for the selection of attributes. They all make sense, but (especially as we get into higher-level attributes) there's no reason to believe they're extensive.
>
> We broadly include many attributes (13 image attributes + classes + scenes + GANalyze) for analysis. But of course, some attributes are reflecting basic statistics and some are human-designed attributes reflecting human perception. That’s why we further use GANalyze to try to discover something that predefined attributes cannot find. We firmly believe that there are some trends that cannot be well summarized or perceived by humans but may still exist. It would be super interesting if we can find a way to discover them further.
>
> Q4: The most notable example is of classes - in table 1, we don't see standard deviation.
>
> We have added the standard deviation to the ImageNet classes with more analysis on the numbers to the revised manuscript. Overall, for top classes, the standard deviations are rather small, mostly between 0.024 to 0.033. For bottom classes, the standard deviation could go higher, ranging from 0.03 to 0.057. For reference, the average standard deviation of all classes is 0.046, which should be considered as a small number. The standard deviation numbers should provide stronger evidence to support our claims.
>
> Q5: Furthermore, the paper claims that classes with multiple objects tend to be high, but previously we were told number of objects was only weakly correlated with MachineMem and that it was not a notable factor of memorability.
>
> In Figure 1, the spearman correlation between number of objects and MachineMem score is 0.09 so we state it is weakly correlated. In GANalyze, the network manipulates the image to have more or less MachineMem scores, and the manipulation strength is quite high, so it enlarges such trends. We have made necessary modifications to make it more clear and self-consistent in the revised manuscript.
>
> Q6: Claims about informative viewing angles, irregular shapes, and "appetizing" food (defined how?) rest purely on examples
>
> Yes, there are some new trends we observed through GANalyze, and they are very hard to give a formal definition with quantitative numbers. They are mostly based on human’s common experience to form a trend.
>
> Q7: Figure 8 hardly backs up the claims made about it. The idea that "machines within the same category generally exhibit strong correlations" is only true for modern CNNs.
>
> There is 1 outlier (AlexNet and MaxViT) that is against the conclusion. So, when we draw conclusions, we state it generally. We have made necessary modifications to the statement in the revised manuscript to make it clearer and acknowledge the outlier case.

---

> > ### Author Response · Authors · 2024-05-06
> > **More responses**
> >
> > Q8: Figure 9 is also hard to draw conclusions from.
> >
> > Our conclusion is that the memory capabilities of a ResNet-50 model are not significantly influenced by its pre-training knowledge (average ρ = 0.35). Stating significantly influenced or not may depend on how to interpret this 0.35 number. It might be explained as a weak to moderate level correlation, so it is indeed more precise to state that different pre-training methods or knowledge will influence the memory capabilities of the same model, but there still exist some moderate correlations that show that machines, regardless of pre-training knowledge, may still tend to memorize similar images in a moderate manner. We have made necessary modifications to this conclusion  in the revised manuscript.
> >
> > Q9: From what I understand, stage (c) doesn't involve previously unseen images. Why not? How would we reason about unseen images being predicted as "seen"? Is that not an important aspect of memorability? If I'm wrong about stage (c) not considering previously unseen images, why are those involved in the analysis?
> >
> > Stage (c) does not involve unseen images since we follow the definition of the visual memory game, and we can only collect the numbers for those seen images (if we follow the visual memory game). But we do agree unseen images are also very important, so we  include them in a variant for calculating the calibration score. And our conclusion is that unseen image sets for calculating calibration tend to behave similarly without including them. This suggests that our pipeline also has a good understanding of unseen images where such unseen images are usually predicted as unseen as expected with low calibration errors.
> >
> > Q10: What makes modern CNNs a distinct category from classic CNNs? The residual objective?
> >
> > Yes, we also believe that’s the main difference!
> >
> > Q11: Presentation
> >
> > We have made all the suggested modifications to the manuscript.
> >
> > Requested changes:
> >
> > R1: More structured writing.
> >
> > We have made the suggested modifications to the manuscript. For example, we have moved some less important sections to the appendix. We have added more structured paragraphs as well as summaries of contributions in the abstract.
> >
> > R2: Stronger descriptive statistics and specific backing for claims.
> >
> > We have modified some of our claims to ensure all claims are supported with stronger evidence. Please feel free to check if any claims are not fully supported, and we are more than happy to adjust further.

---

> > > ### Comment · Reviewer_RWb5 · 2024-05-23
> > > **Con't**
> > >
> > > Q8: "There still exist some moderate correlations that show that machines, regardless of pre-training knowledge, may still tend to memorize similar images in a moderate manner" - where? The edits only introduce more vagueness.
> > >
> > > Q9: Helpful!
> > >
> > > Q10: In that case it would be much more clear to just label them "residual" rather than using an evocative but imprecise name like "modern".
> > >
> > > Q11: Awesome!
> > >
> > > Overall: I really appreciate the careful attention to the review, and I do think the paper's clarity has improved a lot. However, the (lack of) support for the central claims still exists. The correlations that are being distinguished, claims that used to be contradictory have been resolved by weakening them, etc. There needs to be much more experimentation that convinces us of what's memorable and what isn't, and that MachineMem is meaningful for model evaluation/interpretation/other analysis.

---

> > > > ### Author Response · Authors · 2024-05-24
> > > > **Thank you and discussions**
> > > >
> > > > Thank you for your response and for your insightful comments. We appreciate the opportunity to further discuss the paper!
> > > >
> > > > Here we would like to discuss the raised questions further with you:
> > > >
> > > > Q1: In terms of non-ML metrics, (pixel) value of images could be a good measure. This is demonstrated in the top right corner of Fig 4, where the Spearman correlation between value and MachineMem score is -0.40. This is the strongest trend among all 13 metrics we've studied, providing interpretable evidence that images with light backgrounds tend to have low MachineMem scores. The Spearman correlation between contrast and MachineMem score is 0.33 (also shown in Fig 4), further supporting our claim about low MachineMem images lacking contrast or having light color backgrounds.
> > > >
> > > > Q2: We set the threshold for no correlation and very weak correlation at 0.08 (as described in the caption of Fig 4), following common practice that typically sets the range for “no correlation” at 0.1. While there isn't a significant difference between 0.04 and 0.10, reasonable differences do exist. For instance, in Fig 4, the curve of Colourfulness (with a 0.04 correlation) drops four times among 10 groups, but for the curve of Entropy, this only happens once. Our intention was not to rationalize any trends, but to provide precise descriptions and summaries of all trends according to a reasonable standard. The term "overall" in our statement “Overall, the trend xxx…” is merely an introduction to our conclusions.
> > > >
> > > > Q3: Thanks for your further insightful comments on this point!
> > > >
> > > > Q7: The interpretation of "generally" may vary. This statement holds true for modern CNNs (or residual CNNs), and conventional machines, where LC and SVM have a much stronger correlation than LC/SVM with other machines. If we disregard the outlier between AlexNet and MaxViT, this statement still holds for the remaining categories. For classic CNNs, AlexNet and VGG share the strongest correlation, and for ViTs, XCiT and MaxViT share the strongest correlation. Therefore, for four categories, two are always true, and the remaining two depend. We believe "generally" could be a suitable description for this case, but to avoid ambiguity, we have replaced it with more precise numbers.

---

> > > > > ### Author Response · Authors · 2024-05-24
> > > > > **More discussions**
> > > > >
> > > > > Q8: Please refer to page 12, where the revised conclusion is located just above Section 8 Discussion. We've also included the edited text here: "In general, the memory capabilities of a ResNet-50 model are influenced by its pre-training knowledge. However, moderate correlations (average p = 0.35) are still present, indicating that machines, regardless of pre-training knowledge, tend to memorize similar images moderately. This could be because, although pre-training tasks are designed differently, they all aim to learn a strong and transferable representation. As a result, they are expected to share some similarities in terms of memory characteristics." Please let us know if you find this conclusion vague. For further discussions on this statement, you might also want to check our discussions with Reviewer HDw4 on the point R5 “I don't see sufficient evidence, in terms of the moderate correlations observed, to support the conclusion (stated a bit unconvincingly): "regardless of pre-training knowledge, tend to memorize similar images in a moderate manner."”
> > > > >
> > > > > Q10: Indeed, we have made the necessary modifications!
> > > > >
> > > > > Overall comments: We would like to express our gratitude once again for your insightful and helpful comments, which have significantly improved our manuscript. We have conducted preliminary explorations on additional experiments and the practical applications of MachineMem scores (as discussed with Reviewer w2F4 and added to the revised manuscript). For instance, in the context of data augmentation, the MachineMem score without augmentation is 0.678. However, with the application of widely used augmentation methods such as AutoAug and ColorJittering, the scores improve to 0.691 and 0.685 respectively. This indicates that data augmentation can indeed influence machine memorability scores. While it might be beneficial to adopt other memorization metrics, using machine memorability scores could potentially be more appropriate since data augmentation should be developed for machines, not humans.
> > > > >
> > > > > As suggested for additional experiments, another compelling practical application of MachineMem scores is in identifying dataset bias [1]. This paper basically demonstrates that machines can be easily trained to discern the dataset from which images originate. We randomly sampled 100,000 images from each dataset, including ImageNet, LAION, and YFCC. The mean MachineMem scores for these datasets are 0.671, 0.662, and 0.654 respectively. This variance clearly suggests that MachineMem scores could serve as a valuable tool in investigating dataset bias.
> > > > >
> > > > > There are indeed more research questions that could incorporate MachineMem scores. We believe that conducting experiments to show results in two directions should be sufficient to demonstrate the potential of MachineMem scores in the machine learning and computer vision community.
> > > > >
> > > > > [1] A Decade's Battle on Dataset Bias: Are We There Yet? Zhuang Liu and Kaiming He. arXiv 2024.

---

> ### Comment · Reviewer_RWb5 · 2024-05-23
> **Follow-up to justify decision**
>
> Thanks for the responses!
>
> Q1: The analysis is helpful, but those scores are quite similar. It takes away from the result. I also wonder if you could use simpler statistics that don't involve training an ML model so that they would be more interpretable.
>
> Q2: I feel like this is still contradictory, and either way not a significant difference. 0.04 and 0.10 are both considered "weak" in the paper even if they aren't close to each other (a statement I'm not sure I agree with when the scale is 0-1). Now changing that to "no" and "very weak" feels like rationalizing, and it doesn't change the fact that these two weak results are treated differently. Is 0.04 vs 0.10 correlation with one group really a huge difference? And what do you mean by "overall trends"?
>
> Q3: I agree. This doesn't excuse the fact that the high-level attributes aren't extesnive, but I get that it's sort of impossible to be complete.
>
> Q4: Helpful!
>
> Q5: Thanks for the clarification.
>
> Q6: Thanks for the clarification.
>
> Q7: The issue is that the big claims aren't that well supported by the graphs as a whole; it wasn't just AlexNet and MaxViT. What about the fact that "machines within the same category generally exhibit strong correlations" is actually generally *false*?

---

### Review · Reviewer_HDw4 · 2024-04-20

**Summary Of Contributions:**

The authors study a notion of what makes an image more/less memorable to a "machine" -- along the lines of prior studies of a similar notion for humans. The machines in question can be broadly defined as learning algorithms with pre-training objective drawn from a set of self-supervised tasks followed by training for the classification task of whether the image was seen before or not. The authors begin by defining the machine memorability score before reporting this number for a representative machine. The prediction of this score for a given image is also studied as a standalone regression problem. The authors then proceed to study which aspects make an image more or less memorable, highlighting that unlike humans who seem to remember images based on e.g. semantics, what matters for machines seemed to be pixel statistics. Finally, the authors compare different implementations of machines claiming the "memorability" of a given image is an intrinsic property of the image itself that seems to hold across different (or all?) machines.

**Audience:**

Yes

**Broader Impact Concerns:**

The paper makes a very broad claim about an intrinsic image properties with potential impact on many AI/ML algorithms and systems. Before I'm comfortable recommending such a paper for acceptance, I'd like to see a very clear presentation of the methodology and contributions, matched with a solid execution that convinces of all key claims entailed in those contributions. The current draft falls short of this, and I provided some comments based on what I can see to help make progress on those gaps.

**Claims And Evidence:**

No

**Requested Changes:**

- Introduction
  - The introduction is noticeably shorter than what I'm used to. It quickly goes into a brief summary of the study itself, without first providing enough context from the broader area this work belongs to.
  - The first paragraph is a bit too generic, leaving the motivation to study memory unclear. (R1) Please rewrite, putting more emphasis on the role of memory in intelligence. It would be nice to also clarify the potential implications of this study for practical applications of AI/ML and future research in general.
  - (R2) The term "machine" needs to be defined early on with adequate discussion and justification. It would also help to justify the choice of this word compared to other options such as "algorithm" or "program." (I can see in S3 that the paper ends up relying on a particular learning paradigm, see below.)
  - In concluding the introduction, it would help to (R3) summarize the contributions preferably as a clear list. It is also typical to give an overview of the structure of the paper.

- S3
  - (R4) Recommend to add an opening paragraph clarifying the goal and contents of this section, in relation to the rest of the paper.
  - (R5) Please justify the choice to implement machines using this particular learning paradigm.
  - (R6) Please discuss alternative implementations and their relative pros and cons to the one chosen. For one thing, I imagine compressive sensing could have been used to progressively store observations in a compact form and attempt to measure reconstruction errors of new observations wrt the "learned" basis as a measure of memory recall.
  - (R7) Please clarify the details of computing the calibration error, possibly in an appendix.
  - (R8) Please rephrase to clarify "The interchange between stage (b) and stage (c) is iterated"
  - (R9) Please justify choosing the configuration with the lowest calibration error.
  - (R10) Please justify the claim that this approach captures both short- and long-term memory.

I'm leaning to hold off on critiquing the rest of the technical sections, but should be able to take a second look after a revision.

Nitpicking
- P2, 2nd paragraph: recommend to replace "Armed" with a simple technical term such as "Equipped" or just "Using." (The word armed typically brings to mind weapons.)
- P2, 1st paragraph in S2: recommend to replace "preserve," perhaps "propose" or "coin."
- P4, below Eq(1): recommend to replace "force," perhaps "ensure." Please clarify if this required more than simply training for 60 epochs.

**Strengths And Weaknesses:**

For strengths, I can note the following:
- The methodology, which seems to follow prior related works.
- The claim that machine memorability is an intrinsic property of the image itself is very interesting. However, the state of the draft falls short of convincing me to accept this - at least on the first reading. The authors are encouraged to do their part to establish this claim, see below for some suggestions.

At a high level, the following aspects need attention:
- Clearly state the following in the introduction: goals, key questions, methodology, and contributions.
- Justify the choice of the primary machine used in sections 3 and 4. It was good to see that eventually other choices were compared.
- The paper tackles a relatively large number of questions, and the overall quality of execution seems to have suffered as a result. It would help to identify 1-3 key questions and make a more focused story around those, possibly deferring remaining/secondary questions to a follow up paper or an appendix.

---

> ### Author Response · Authors · 2024-05-06
> **Thank you for your comments**
>
> Thank you for your insightful comments on our manuscript. Your suggestions have greatly improved our work.
>
> Questions:
>
> Q1: Clearly state the following in the introduction: goals, key questions, methodology, and contributions.
>
> We have rewritten the introduction to make it clearer. Also, a summary of novel contributions has been added in the revised manuscript.
>
> Q2: Justify the choice of the primary machine used.
>
> We use the most widely used ResNet-50 as the primary machine given its widespread use and strong generalization in various visual tasks. We have included our justifications in the revised manuscript.
>
> Q3: The paper tackles a relatively large number of questions, and the overall quality of execution seems to have suffered as a result. It would help to identify 1-3 key questions.
>
> We have moved some less important content (section 5.3 What scenes are more or less memorable? And 5.4 Can more or less memorable classes be semantically grouped according to a hierarchical structure?) to the appendix to make the main paper more concise with a clearer story.
>
> Requested changes:
>
> R1-R3: Introduction
>
> We have rewritten the introduction part following the requested changes.
>
> R4-R10: Section 3
>
> We have adjusted the manuscript according to your very valuable suggestions.
>
> Nitpicking:
>
> Thanks for nitpicking! They are very valuable. We have made corresponding changes.

---

> ### Comment · Reviewer_HDw4 · 2024-05-17
> **Comments on revision**
>
> Thanks for incorporating previous comments/requests and engaging with the reviewers.
>
> Regarding prior requests:
> - (R5) really asks to justify the choice of this self-supervised learning paradigm, per Eq.1, as the primary machine rather than other alternatives such as the ones considered later in Section 7.2. Indeed, this section concludes that, for a fixed architecture, the choice of the pre-text task influences the memorability score.
>   - I don't see sufficient evidence, in terms of the moderate correlations observed, to support the conclusion (stated a bit unconvincingly): "regardless of pre-training knowledge, tend to memorize similar images in a moderate manner."
>
> Some further comments on the revised manuscript:
> - Introduction
>   - Second paragraph starts off with a circular definition of memory. Please (R11) provide a clear technical and usable definition of what's meant by memory at least within the scope of the paper. Memory immediately brings to mind RAM or disk, with the primary tasks of storage and retrieval as traditionally achieved by various techniques from data structures, databases, or information retrieval systems. It is not immediately clear what memory means for a pattern recognition system, i.e., learning-based classifier.
>   - The adoption and adaptation of human memorability scoring needs more justification. It may help to recast human memorability testing as a "guessing game," of whether an image was seen before or not, in order to frame it as a classification task and distinguish that from retrieval. It could also help to contrast this image-based game to text/speech, where the participants could be asked to "recite" one of a number of texts/recordings they were given, as an example of retrieval from memory.
> - Section 6:
>   - Please (R12) use a more descriptive title for Section 6.3.
>
> Overall assessment:
> - I see some merit in the results presented in Section 6.1. I'm still not convinced with the interpretation of the results, given the concerns I raised above regarding the basic definitions and methodology of the study.
> - I would be more comfortable recommending this article for publication under a different title that better captures the questions, methodologies, and results presented. Perhaps something like: "On a notion of image memorability under self-supervised pre-training"

---

> > ### Author Response · Authors · 2024-05-17
> > **Thank you for your further comments**
> >
> > Thank you for getting back to our response and raising new valuable suggestions!
> >
> > R5: In designing the MachineMem measurer, our aim is to replicate the visual memory game used to measure human memorability of images as closely as possible. In this game, humans view sequences of images, with each image presented individually and without any distortions. Our design principle led us to use the rotation prediction task (Eq.1) in stage (a) for the following reasons: it is the closest method to replicating the visual memory game, mirroring how humans view images. An added advantage is that this rotation prediction task does not necessitate any architectural modifications within machines/networks (except for the final linear layer, which is not part of the representation), allowing us to directly use different pre-trained weights as the initialization to conduct the study in Section 7.2.
> >
> > We conducted the study in Section 7.2 with the understanding that humans, regardless of their background, tend to memorize similar images. We wanted to see if machines exhibit a similar trend, treating the different knowledge learned from various pre-text tasks as the background. We used the ResNet-50 weights from different pre-text tasks as the initialization and ran the MachineMem measurer process for each one to conduct the study.
> > Although pre-text tasks are designed differently, they all aim to learn a strong and transferable representation. Therefore, it is not surprising to find that machine memorability scores collected from them share some correlations (average p=0.35). We hope this provides sufficient evidence to conclude that "regardless of pre-training knowledge, machines tend to memorize similar images in a moderate manner." We have added our explanations and analysis regarding representation learning to the revised paper.
> >
> > R11 Definition of machine memory: We have revised the second paragraph to provide a description of memory in machines within the context of this paper.
> >
> > R11 More justification on visual memory game adoption: The visual memory game is essentially a "guessing game". Participants in this game only need to indicate whether they have seen an image before or not, making it naturally adaptable to stage (b) of our MachineMem Measurer. Furthermore, such a visual memory game can be extended to other signals. For instance, [1] demonstrates that the most memorable words have a one-to-one relationship with their meanings, which aligns with the fact that humans tend to memorize "simple" images. The talk [2] starting from 16:45 also covers some content on the memorability of words. We have added more clarifications on the visual memory game to the preliminary section to avoid confusion.
> >
> > [1]:  Intrinsically memorable words have unique associations with their meanings
> >
> > Greta Tuckute,  Kyle Mahowald,  Phillip Isola,  Aude  Oliva, Edward Gibson, and EvelinaFedorenko.
> >
> > [2]: https://www.youtube.com/watch?v=43kansULeBE
> >
> > R12: We have updated the title to be more informative.
> >
> > Paper title: Indeed, this paper focuses on studying the memorability of images with regard to pattern recognition machines. Would it be helpful to incorporate this constraint into the title, resulting in a new title such as 'On the notion of image memorability for pattern recognition machines'? For self-supervised learning, however, we feel it might be better to exclude it since we also perform lots of experiments on machines trained from scratch.
> >
> > Thank you once again for your valuable comments and insightful discussions. We have uploaded a revised version, highlighted in blue, to cover the requested changes.

---

> ### Comment · Reviewer_HDw4 · 2024-05-22
> **Response to second round of comments**
>
> Thanks for your patience following up on reviewer discussions.
>
> I'm afraid I still don't think we have a satisfactory resolution to R5. The latest comments form the authors exhibit deeper confusions and/or controversies:
> -  Excerpt-A: *"Our design principle led us to use the rotation prediction task (Eq.1) in stage (a) for the following reasons: it is the closest method to replicating the visual memory game, mirroring how humans view images. An added advantage is that this rotation prediction task does not necessitate any architectural modifications within machines/networks (except for the final linear layer, which is not part of the representation), allowing us to directly use different pre-trained weights as the initialization to conduct the study in Section 7.2."*
>   - Response: I find it difficult to see how rotation prediction can be related in any manner to either (1) how humans view images, or (2) the visual memory game itself. The added advantage, or perhaps convenience, mentioned later, doesn't merit using this method as the basis for this study.
> - Excerpt-B: *"Although pre-text tasks are designed differently, they all aim to learn a strong and transferable representation."*
>   - Response: I tend to believe that pre-text tasks, by definition, aim to solve the pre-text task. Whether or not they learn effective features for other tasks depends on many factors. The degree of correlation shown is interesting, but given the issues and confusions with the methodology discussed above, it's not clear how to interpret those results.
>
> The proposed title would be better, though I don't see it reflected in the submission.

---

> ### Author Response · Authors · 2024-05-23
> **Follow-up**
>
> We would like to express our gratitude once again for your consistent follow-ups, patience, and valuable suggestions.
>
> We would like to provide further clarifications regarding your concerns.
>
> Excerpt-A: In the visual memory game, the images sent to humans possess two properties: they are natural and sent one-by-one. For the first property, we aim to make as minimal changes as possible to the images sent to machines. Rotation is a suitable choice in this context as it doesn't significantly alter the overall meaning or semantics of the images. If we opt for masked image modelling or inpainting, it would introduce too many corruptions to the images and would not align with the visual memory game. It would also necessitate numerous additional modifications to the network architecture, which is not conducive to our study, as in stages (b) and (c), we need to keep the weights learned from stage (a). For the second property, where only one image is sent to humans at a time, rotation prediction is also a suitable task. If we choose contrastive learning, it might require sending multiple image instances to the network, deviating significantly from the visual memory game.
>
> Excerpt-B: We concur that pre-text tasks are designed to solve the pre-text task itself. However, if the pre-text task requires strong feature extraction abilities, which is true for many self-supervised learning pre-text tasks, the network will naturally learn how to encode images and extract low-level features in the shallow and middle layers [1, 2]. The deeper layers will tend to be more task-specific. In this sense, the shallow and middle layers should naturally share some similarities. This is reflected in the degree of correlation regarding machine memorability scores.
>
> Title: Thank you for confirming our proposal! We have updated the paper title accordingly.
>
> Thank you once again for your continued support and engagement in this discussion.
>
> [1]:  How transferable are features in deep neural networks? Jason Yosinski, Jeff Clune, Yoshua Bengio, Hod Lipson. NeurIPS  2014
>
> [2] A critical analysis of self-supervision, or what we can learn from a single image. Yuki M. Asano, Christian Rupprecht, Andrea Vedaldi. ICLR 2020

---

### Review · Reviewer_w2F4 · 2024-04-22

**Summary Of Contributions:**

This paper has two core contributions; first, it proposes an ML analog to human visual memory games, and analyses models using this analog; second, it uses those scores to train a predictor of those scores, and using this predict performs a wide array of empirical work.

The human visual memory game consists in showing a series of images to humans, asking them to identify already seen images as they are presented. The score in this game is simply the percentage success of the human.

Since a typical model does not have short term memory*, this task is adapted to ML into three stages. First the model is finetuned on a self-supervised task (predict rotation), then it is again finetuned to predict whether the input was in the first stage, it is then tested on first stage images that were held out in the second stage.

This is repeated a number of times, on the same images to assign them a memorability score, creating a dataset on which another predictor model can be trained.

This trained predictor is then used to comment on which types of images tend to be more memorable according to this metric.

*This is not entirely true anymore, considering the advent of large language models with amazingly large context windows.

**Audience:**

No

**Claims And Evidence:**

No

**Requested Changes:**

I think what would make this paper more complete is to show one of several things:
- that the proposed metric is more interpretable than much simpler memorization metrics (e.g. that of Feldman et al.), e.g. in some of the correlations measured
- that the proposed metric is actually useful in downstream tasks such as data augmentation
- a much richer comparison with past work; how is this different from past attempts to understand memorization in ML?

The presentation could be improved.
- The text assumes a lot from the reader, and many details are left unsaid. What it means for something to be memorable is basically not explained until the middle of page 5. What is the visual memory game? How are humans normally presented the images? Why would we expect machines to behave similarly? None of this is explained or justified. This makes it hard to understand the motivation of this paper as one reads it.
- Citations should be properly formatted.
- It should be a bit clearer what is a novel contribution and what isn't; a lot of prior work, analysis setups and tools are reused in this paper. Doing so is fine but it's better when what is novel vs what is reusing prior work is delineated properly.

**Strengths And Weaknesses:**

The main strengths of the paper is that the empirical investigation is fairly complete. Many models were tried and compared and many experiments were run. The topic is also an interesting one, understanding how deep models work is important.

That being said, I ask myself, "what did I learn from the paper?" and cannot form a good answer.

Let's start with the analog task. I'm not sure what running it shows, other than something I believe to be already well known, which is that large enough models excel at few-shot learning, and more generally, are easily able to be adapted to a new similar task. What's complicating this analysis is that it's also entangled with forgetting. Are the models forgetting set A during phase 2? Or did they just fail to capture set A at all in phase 1?
It's also well known that there are examples which are harder than other examples to fit for neural networks, this is even observable in the Fourier domain (i.e. what is mainly seen in Figure 3). That there is variance in the analog task should be unsurprising.

Comparing to humans and human scores seems very strange. There seems to be a presupposition that permeates the text, that somehow we should expect humans and machine to think similarly. This seems to be both a factual claim (e.g. we should expect the same images to be memorable to both) and an esthetic claim (that it is desirable for machines to think like humans). Setting aside the esthetics, there are many reasons to believe that we should not in fact expect humans and deep models to look similar. Even similarities between the visual cortex and convolutional models, perhaps the two systems that are the closest in form and function, should generally be avoided except in specific settings [1].

Specifically here, I'm not sure why this analog task is developed. Even if we were aiming for human-like reasoning, with current architectures the test ends up being very different from how humans would presumably perform it internally (which we know very little about). In terms of what it informs us about ML, the conjectures in the text are also fairly unsatisfying: "we might design a new data augmentation strategy that can make data more memorable to machines to assist the training of neural network." This supposes lots of things about memorable images which we have very little indication are true. Why not use simpler memorization metrics?

Finally, I'm very wary of sections 4 and onwards, for the simple reason that there's _a lot_ of bootstrapping going on. In the experiments where the score predictor and GANalyze is used, there are so many steps where approximation is used that I find it really hard to trust the conclusions drawn in the paper. A model is trained to predict a model of memorization, and is then used adversarially in a generative model to generate approximate inputs of what could be memorable. There are many steps where something could go wrong, not least that inputs generated by GANs are unlikely to look anything like natural training data.

[1] Barrett, David GT, Ari S. Morcos, and Jakob H. Macke. "Analyzing biological and artificial neural networks: challenges with opportunities for synergy?." Current opinion in neurobiology 55 (2019): 55-64.

---

> ### Author Response · Authors · 2024-05-06
> **Thank you for your comments**
>
> Your constructive feedback on our manuscript is greatly appreciated. Thank you for helping us enhance our work.
>
> Questions:
>
> Q1:  What did I learn from the paper?
>
> The main conclusion we want to present is that machines tend to memorize 'complex' images, which is very different from humans, who tend to memorize 'simple' images.
>
> Q2: What's complicating this analysis is that it's also entangled with forgetting. Are the models forgetting set A during phase 2? Or did they just fail to capture set A at all in phase 1?
>
> Set A does capture images from set A. We conducted a simple experiment to verify this. We collected a new set D (all unseen), distinct from A, B, C. When we performed stage C on this new set D, more than 84% of images inside D were classified as unseen. However, for set A, the usual seen rate is more than 60% (which means less than 40% are classified as unseen). This significant difference should be able to show that Stage A does capture images from set A, further verifying our setting to be reasonable.
>
> Q3: Comparing humans and human scores seems very strange. There seems to be a presupposition that permeates the text, that somehow we should expect humans and machines to think similarly.
>
> Though neural networks in the machine learning community are designed to imitate human neurons and are expected to behave similarly to humans, we all know this is not commonly happening and machines can behave very differently from humans. So, a long-term yet very important goal of the machine learning community is still to develop machines that may tend to behave more like humans. To do this, it first needs to have a very deep understanding of their similarities and differences. We hope to use memorability to shed light on this, where machine memorability score is a topic that has never been touched before. We firmly believe the comparison between humans and machines should provide valuable trends and insights; this is also appreciated by other reviewers.
>
> Q4: In terms of what it informs us about ML, the conjectures in the text are also fairly unsatisfying: "we might design a new data augmentation strategy that can make data more memorable to machines to assist the training of neural networks."
>
> Conjectures could be proven to be true or false, we make this conjecture based on our understanding and hope it could be verified in future works. Designing a new augmentation method is beyond the scope of our current work and we hope to leave it for future explorations. However, we have added some new experiments to provide stronger evidence to support our conjecture. In this experiment, we measure the machine memorability scores of different augmentation methods applied to the same image set. The MachineMem score without augmentation is 0.678, but with the application of widely used augmentation methods such as AutoAug and ColorJittering, the scores improve to 0.691 and 0.685 respectively. This suggests that data augmentation can indeed impact machine memorability scores. We have added this to the revised manuscript.  It might also be good to adopt other memorization metrics, but as data augmentation should be developed for machines not humans, using machine memorability scores could potentially be more reasonable.
>
> It might also be good to adopt other memorization metrics, but as data augmentation should be developed for machines not humans, using machine memorability scores could potentially be more reasonable.
>
> Q5:In the experiments where the score predictor and GANalyze is used, there are so many steps where approximation is used that I find it really hard to trust the conclusions drawn in the paper.
>
> We agree GANalyze might not be the best way to show such trends of memorability scores in the future, but at the current stage, we fairly believe it is the most suitable way. GANs are trying to generate images following a given distribution, and it is a well-developed way to do so. Also, GANalyze, published at ICCV 2019 is recognized as a solid paper in the field, which already draws more than 300 citations (given its focus on studying image properties which is not a hot topic.). More importantly,  GANalyze was mainly developed for manipulating human memorability scores, which is a perfect match for our study. We fairly believe that it could be a good way to show how images will look like according to the changes in machine memorability score. Anyway, we feel that there is no better way to do it and GANalyze is already the best choice that can possibly achieve. The results presented through GANalyze are appreciated by reviewer RWb5 and we believe it would be valuable to keep them.

---

> > ### Author Response · Authors · 2024-05-06
> > **More responses**
> >
> > Requested changes:
> >
> > R1:The proposed metric is more interpretable than much simpler memorization metrics (e.g. that of Feldman et al.).
> >
> > Could you please specify the full bib of Feldman et al?
> >
> > R2: That the proposed metric is actually useful in downstream tasks such as data augmentation
> >
> > Please refer to Q4.
> >
> > R3: A much richer comparison with past work; how is this different from past attempts to understand memorization in ML?
> >
> > The main difference is that we are studying machine memorability of images, which is a new concept that has not been studied before, and could provide a new perspective for the machine learning community to look at memorization and memorability problems. We have discussed Memory modules and Memorization in DNNs in related work, and we have further highlighted the differences in our revised manuscript.
> >
> >
> > R4: The presentation could be improved.  The text assumes a lot from the reader, and many details are left unsaid. What it means for something to be memorable is basically not explained until the middle of page 5.
> >
> > We agree that it could be beneficial to present more details on the visual memory game as a complement to some general descriptions inside related work. We have added a preliminaries section before the method section to systematically explain the visual memory game and the definition of human memorability scores in the revised manuscript.
> >
> > R5: Citations should be properly formatted.
> >
> > We have checked all citations and re-formatted some of them. They should be presented in a better shape now.
> >
> > R6: It should be a bit clearer what is a novel contribution and what isn't; a lot of prior work, analysis setups and tools are reused in this paper. Doing so is fine but it's better when what is novel vs what is reusing prior work is delineated properly
> >
> > We have added a new paragraph at the end of the introduction to highlight our novel contributions in the revised manuscript.

---

> > > ### Comment · Reviewer_w2F4 · 2024-05-27
> > > **Updates**
> > >
> > > Sorry for the very late engagement--NeurIPS rush. Thanks for all the follow ups and the edits to the paper. I think the edits made the points you're trying to make much more compelling.
> > >
> > > - The current title in the paper has a "Spearman correlations", I'm assuming that's a mistake
> > > - The Feldman paper I was thinking of is the one you cite by Feldman & Zhang
> > > - I think the extra results useful, they at least ground your claims a little more
> > >
> > > I still do think the paper could be empirically stronger, in a sense papers can always be, but this passes the TMLR bar for me.

---

> > > > ### Author Response · Authors · 2024-05-27
> > > > **Thank you and brief response**
> > > >
> > > > Thank you for returning to this discussion after the NeurIPS deadline! We greatly appreciate your engagement, clarifications, and additional comments!
> > > >
> > > > Please allow us to provide a short response:
> > > >
> > > > Title: Thank you for pointing it out. Indeed, it was a mistake, and we have corrected it.
> > > >
> > > > Feldman Paper: Thank you for the clarification. The primary reason we do not use this metric is that we aim to adhere to the definition and measurement of human memorability scores. Feldman et al. primarily focus on label memorization, which is something we intend to eliminate in our study. For more details, please refer to the discussions in the "Memorization in DNNs" section of the Related Work.
> > > >
> > > > Extra Results: We are glad to hear that they are useful. Thank you for your additional feedback.

---

### Comment · Reviewer_Vdt7 · 2024-04-05
**Absolutely not**

Please unassign. I am not available at the moment.

---

### Author Response · Authors · 2024-05-06
**General response**

We would like to express our gratitude to the reviewers for their insightful and constructive comments. We have addressed each reviewer's feedback individually and have uploaded a revised version of the manuscript. The primary modifications in this updated version are highlighted in green, encompassing all the requested changes, unless discussed in individual responses. This revised manuscript should now provide more robust evidence to substantiate our claims. Please feel free to provide any additional comments; we are always open to further discussion. Once again, we thank you for your valuable review!

---

> ### Author Response · Authors · 2024-05-16
> **Reminder**
>
> We would like to express our gratitude once again for the valuable time reviewers have spent on our work and for your insightful suggestions.
>
> This is a friendly reminder that our discussion period will be ending in four days. If you have any additional suggestions, please feel free to share them with us.
>
> We are more than willing to further improve our paper based on any additional suggestions you may have for our updated version.
>
> Best Regards,
> The Authors

---

### Decision · Action_Editor_TEhS · 2024-05-30

**Recommendation:** Reject

**Comment:**

Area chair and reviewers generally agree that the MachineMem score is interesting on its own. It might be an intrinsic property of deep learning models that sheds light on their behavior.

As summarized by the Authors, the main insight derived from the paper is that machines and humans memorize in a different manner.
However, the connection to human perception is in the opinion of the area chair too weak to warrant acceptance at this stage. As Authors confirm, this is the central claim of the paper, and it should be either revised or significantly better supported.

**Audience:**

The paper is clearly interesting to the audience interested in understanding deep learning models. The experiments are generally well-designed and the main insight that the more complex example, the more likely it is to be memorized (as defined by Authors), was appreciated by reviewers and would be likely to be appreciated by the broader audience.

**Claims And Evidence:**

As summarized by the Authors, the main insight derived from the paper is that machines and humans memorize in different manner.

Accordingly, the paper makes three core claims:

1. MachineMem measures faithfully which examples are memorized by deep models (more precisely authors tested only vision classification models)
2. Humans and machines assign scores very differently, in their respective versions of the visual memory game
3. MachineMem and HumanMem both reflect the same characteristic, which in particular — thinking about machine and human and image classifiers —  implies that making MachineMem scores closer to HumanMem score would bring the two systems to operate more similarly in terms of image classification

, where machine memorization is operationalized through a specific training procedure consisting of three stages, and human memorization refers to a standard test in cognitive sciences.

The first two claims are well supported. Area chair and reviewers generally agree that the MachineMem score is interesting on its own. It seems to be an intrinsic property of deep learning models that sheds light on their behavior. The experiments are generally well-designed and the main insight that perceptual example complexity correlates with memorization  (as defined by Authors) was appreciated by reviewers.

However, the third claim is not well supported in the opinion of the area chair, and similar sentiments were shared by two of the reviewers. It is unclear whether the two tasks measure the same thing. The human brain is optimized for many more objectives and constraints than a neural image classifier. Hence, its memorization behaviour is also influenced by these different objectives and constraints it is trying to meet. Conversely, the machines were all optimized to improve image understanding, which might cause them to memorize different examples than humans because of the difference in objectives. Hence, it is unclear if the two systems should have more similar memorization scores or not, given that the human brain is not only optimized for image understanding. Furthermore, it is unclear whether if we aligned more machine memorization scores, they would classify images more like humans or not.

The third claim would be better supported if more causal experiments were designed. For example, can we show that increasing machine memorization scores alignes human and machine perception (e.g. shape vs texture bias)?

**Resubmission Of Major Revision:**

The authors may consider submitting a major revision at a later time.

---

> ### Author Response · Authors · 2024-05-31
> **Thank you for your summary and feedback**
>
> Thank you for your detailed and thoughtful summary and feedback on our paper! We appreciate the time and effort you have taken to review our work.
>
> We are pleased that our first two claims and the MachineMem score were well-received and that our experimental design was appreciated.
>
> Regarding the third claim, we understand the concerns about the alignment and relationship between human and machine memorability scores. We will focus on designing additional experiments to better investigate their relationship.
>
> We are more than willing to revise our work based on your valuable feedback and hope to resubmit a major revision in the near future.
>
> Thank you once again for your constructive comments.